# Inactivating mutations and X-ray crystal structure of the tumor suppressor OPCML reveal cancer-associated functions

James R. Birtley[1,6,7], Mohammad Alomary[2,7], Elisa Zanini[2], Jane Antony [2], Zachary Maben[1], Grant C. Weaver [1], Claudia Von Arx[2], Manuela Mura[2], Aline T. Marinho[2], Haonan Lu [2], Eloise V.N. Morecroft[2,3], Evdoxia Karali[2], Naomi E. Chayen [4], Edward W. Tate [3], Mollie Jurewicz[1], Lawrence J. Stern [1,8], Chiara Recchi [2,8] & Hani Gabra [2,5,8]

*OPCML*, a tumor suppressor gene, is frequently silenced epigenetically in ovarian and other cancers. Here we report, by analysis of databases of tumor sequences, the observation of *OPCML* somatic missense mutations from various tumor types and the impact of these mutations on OPCML function, by solving the X-ray crystal structure of this glycoprotein to 2.65 Å resolution. OPCML consists of an extended arrangement of three immunoglobulin-like domains and homodimerizes via a network of contacts between membrane-distal domains. We report the generation of a panel of OPCML variants with representative clinical mutations and demonstrate clear phenotypic effects in vitro and in vivo including changes to anchorage-independent growth, interaction with activated cognate receptor tyrosine kinases, cellular migration, invasion in vitro and tumor growth in vivo. Our results suggest that clinically occurring somatic missense mutations in OPCML have the potential to contribute to tumorigenesis in a variety of cancers.

[1] Department of Pathology, University of Massachusetts Medical School, Worcester, MA 01655, USA. [2] Ovarian Cancer Action Research Centre, Department of Surgery and Cancer, Imperial College London, Du Cane Road, London W12 0NN, UK. [3] Department of Chemistry, Imperial College London, Wood Lane, London W12 0BZ, UK. [4] Computational and Systems Medicine, Department of Surgery and Cancer, Faculty of Medicine, Imperial College London, London SW7 2AZ, UK. [5] Clinical Discovery Unit, Early Clinical Development, IMED Biotech Unit, AstraZeneca, Cambridge SG8 6HB, UK. [6] Present address: UCB Pharma, Bath Road, Slough SL1 3WE, UK. [7] These authors contributed equally: James R. Birtley, Mohammad Alomary. [8] These authors jointly supervised this work: Lawrence J. Stern, Chiara Recchi, Hani Gabra. Correspondence and requests for materials should be addressed to L.J.S. (email: lawrence. stern@umassmed.edu) or to C.R. (email: c.recchi@imperial.ac.uk) or to H.G. (email: h.gabra@imperial.ac.uk)

Opioid-binding protein cell-adhesion molecule like (OPCML) is a glycosylphosphatidylinositol (GPI)-anchored protein that localizes to the outer leaflet of the plasma membrane and acts as a tumor suppressor (TS)[1,2]. OPCML is a member of the IgLON superfamily, together with 4 other proteins (LSAMP, NEGR1, HNT, and IgLON5), amongst which some have been hypothesized to act as tumor suppressors or oncogenes[3,4]. However, functions of IgLON members have not been fully characterized and their 3D structures remain unsolved. Previously, we showed that OPCML is inactivated predominantly by somatic methylation and loss of heterozygosity in more than 80% of ovarian cancer patients[2]. When re-expressed in cancer cells, OPCML inhibits proliferation in vitro and tumorigenicity in vivo[2] by binding and downregulating a specific subset of receptor tyrosine kinases (RTKs)[5]. Although somatic promoter methylation is the predominant mechanism of silencing in diverse tumor types[1], we previously also reported a somatic missense mutation at P95R in an ovarian cancer patient[2]. Preliminary data indicated that this variant was expressed in similar amounts as wild-type (WT) but was a loss-of-function mutant. Here, we search for additional OPCML somatic mutations in the TCGA and COSMIC DNA sequencing databases from a panel of nearly 30,000 patients affected by different cancers. We show that a number of patients exhibit somatic mutations of *OPCML*. In order to understand how these clinical mutations might interfere with OPCML's tumor suppressor activity we then determine the crystal structure of OPCML and show that these mutations fall into several classes based on mutation type and location on the protein structure. Selecting mutations representative of each class, we characterize their biophysical properties and demonstrate their functionality in in vitro and in vivo models. We further demonstrate that particular mutations affect the oligomeric state of OPCML, its interaction with cognate RTKs and its effect on a range of phenotypes including migration, invasion, adhesion to extracellular matrix proteins and in vivo tumorigenicity. These findings reveal how the mutations result in functional/clinical consequences, and these data have implications for understanding the mechanism of action of this class of membrane-associated tumor suppressors.

## Results

**OPCML sustains point mutations in cancer patients**. We analyzed tumor DNA sequence data from the TCGA and COSMIC databases in order to identify possible clinical *OPCML* mutations. We found point mutations in the *OPCML* gene in 287 out of 28,132 patients and these were distributed across all cancer types (Supplementary Fig. 1A). Generally, the frequency of these clinical mutations is uncommon, ranging from 5.5% in melanoma and 3.3% in colon, gastric, and bladder cancer to around 0.1% in cancers such as breast and brain (Supplementary Fig. 1A). Nine mutations were found to be splice site variants or stop codon-related alterations (Supplementary Table 1). The remaining 278 changes were missense mutations (Supplementary Table 1) and were mostly scattered evenly along the entire OPCML sequence, with the greatest number located in domain 1 (Supplementary Fig. 1B).

**X-ray crystal structure of OPCML**. To localize the clinical mutations on the three-dimensional structure of OPCML and to gain insights into their role in the tumor suppressor mechanism(s) we crystallized soluble recombinant OPCML (residues 36–316) and determined its structure using a combined single-wavelength X-ray anomalous dispersion/molecular replacement approach (Table 1). Structure solution was complicated initially by a high degree of translational pseudosymmetry, but identification of the correct space group and non-crystallographic symmetry elements

allowed for chain tracing, which was followed by rounds of automated and manual model building and refinement (see Methods). In the final model (PDB entry 5UV6), 273 out of 286 residues in the mature OPCML protein were visualized (Supplementary Figs. 2 and 3), with two molecules in the crystallographic asymmetric unit. There are six potential N-linked glycosylation sites (asparagines 44, 70, 140, 285, 293, and 306) and glycans were modelled at residues 70, 293, and 306.

OPCML consists of three Ig-like domains termed D1, D2, and D3, connected by short extended linkers (Fig. 1a, b, Supplementary Fig. 2). The OPCML homodimer resembles an inverted V shape (Fig. 1a), with dimerization mediated by a membrane-distal D1-D1 interface stabilized by a combination of hydrophobic contacts, hydrogen bonds and salt bridges (Fig. 1c, Supplementary Fig. 4). This interface comprises 874 Å$^2$ of buried surface area and has a solvation free energy gain ($\Delta^i$G) on formation calculated by PISA to be $-9.5$ kcal M$^{-1}$. These values are larger than for other intermolecular interaction sites observed in the crystal, the next largest being crystal contacts mediated by the C-terminal end of one molecule inserting between the B and G strands of D1 in another with 554 Å$^2$ buried surface area and solvation free energy gain of $-6.9$ kcal M$^{-1}$ (Supplementary table 2 and Supplementary Fig. 5). Based on the quaternary structure seen in the asymmetric unit, the dimer would be anchored into the plasma membrane by two D3-linked GPI anchors (Fig. 1a). The D3 C-terminal ends are located approximately 180 Å apart, and thus the top of the D1 dimerization interface could extend above the plane of the plasma membrane by ~73 Å (Fig. 1a).

The OPCML dimer is stabilized by reciprocal salt bridges between Arg 65 of one monomer and Asp 80 of the other (Fig. 1c,

**Table 1 Data collection, phasing, and refinement statistics for OPCML (MR-SAD)**

|  | Native | K$_2$PtCl$_4$ |
|---|---|---|
| *Data collection* |  |  |
| Space group | P4$_1$2$_1$2 | I4$_1$22 |
| Cell dimensions |  |  |
| *a, b, c* (Å) | 93.6, 93.6, 262.2 | 98.8, 98.8, 267.1 |
| *α, β, γ* (°) | 90, 90, 90 | 90, 90, 90 |
| Resolution (Å) | 18.5–2.65 (2.74–2.65)[a] | 48.3–3.17 (3.25–3.17) |
| $R_{sym}$ | 17.1 (273.6) | 18.7 (301.5) |
| $R_{pim}$ | 4.7 (73.8) | 3.6 (56.6) |
| $I/\sigma(I)$ | 9.7 (1.2) | 11.5 (1.1) |
| $CC_{1/2}$ | 100 (68.5) | 100 (82.3) |
| Completeness (%) | 95.8 (88.8) | 100.0 (100.0) |
| Redundancy | 14.4 (14.5) | 28.2 (29.1) |
| *Refinement* |  |  |
| Resolution (Å) | 18.5–2.65 (2.74–2.65) |  |
| No. reflections | 33374 (3021) |  |
| $R_{work}/R_{free}$ | 28.7/30.4 |  |
| No. atoms | 4392 |  |
| Protein | 4216 |  |
| Ligand/ion (carbohydrate) | 134 |  |
| Water | 42 |  |
| *B factors* |  |  |
| Protein | 84.6 |  |
| Ligand/ion | 91.0 |  |
| Water | 74.2 |  |
| *R.m.s deviations* |  |  |
| Bond lengths (Å) | 0.006 |  |
| Bond angles (°) | 0.99 |  |
| *NCS RMSD (Å)* |  |  |
| All domains | 1.10 |  |
| D1 (43–134) | 0.24 |  |
| D2 (135–222) | 0.19 |  |
| D3 (223–319) | 0.64 |  |

[a]Values in parentheses are for highest-resolution shell

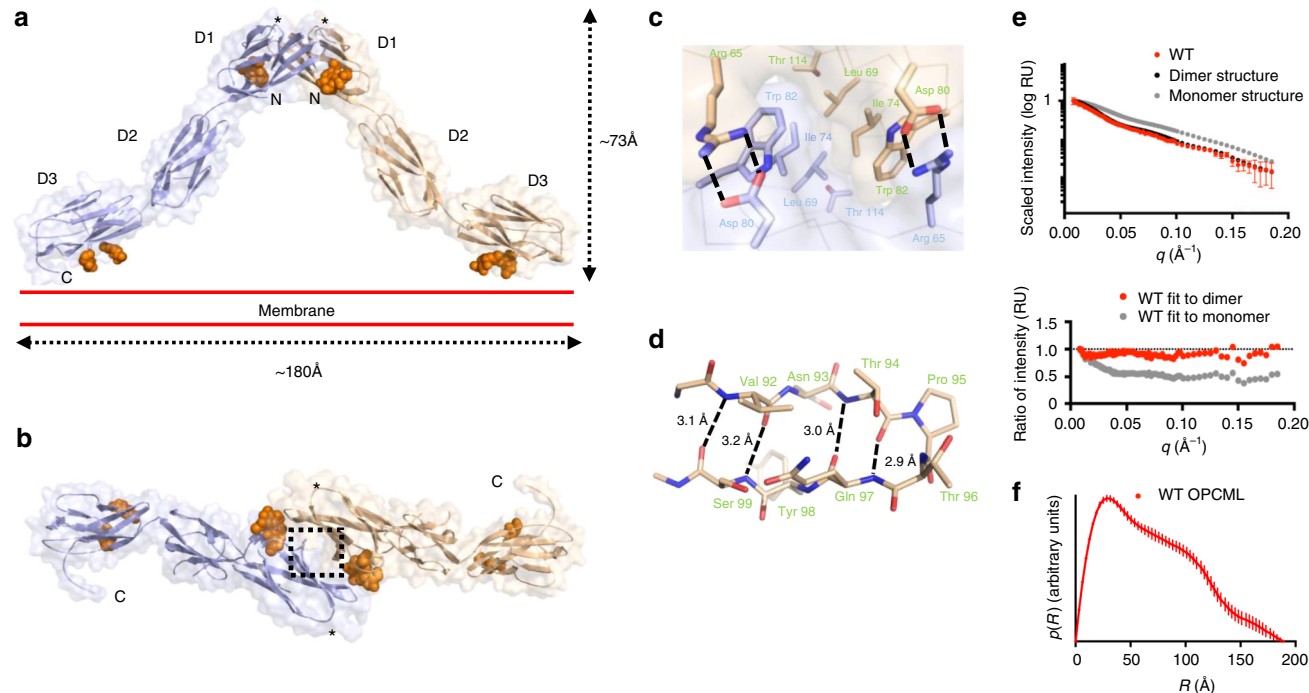

**Fig. 1** Crystal structure of the OPCML homodimer. **a** Ribbon representation overlaid on a transparent surface illustrating the dimeric architecture of OPCML. One monomer is colored blue and the other wheat. N-linked glycosylation (on asparagines 70, 293, and 306) are shown as orange spheres. Location of P95 indicated by asterisk. N- and C- termini are indicated. Red parallel lines show the relative orientation of OPCML to the plasma membrane (on the extracellular side). Scale bars are shown to illustrate maximum dimensions. **b** Represented the same as in **a** but viewed from the top. The dashed rectangle represents key residues at the dimerization interface and these are shown in detail in **c**. **c** The D1-D1 homodimerization interface. Arg 65 from one monomer forms a salt bridge with Asp of the other monomer. Below Arg 65 is a stacking interaction involving Trp 82, Ile 74, Leu 69, and Thr 114. Same color scheme as in **a**, **b** . The alpha carbon backbone is shown as thin lines. **d** Stick representation of the β-turn encompassing residue P95, shown in wheat. Residues 92–94 and 97–99 form part of the D and E β–strands, respectively. Hydrogen bonds are indicated by dashed lines. **e** Small angle X-ray scattering analysis of WT OPCML. Scattering curves were measured and compared to calculated data of the dimer as seen in the crystal structure and of a monomeric model of OPCML. The fit to the dimer ($\chi = 0.50$) matches more closely than to the monomer ($\chi = 2.35$). **f** Pair-distance distribution function curve for WT OPCML. The curve intercepts the x-axis at $R = 189$ Å, indicating the maximum atomic distance in a single scattering particle. This model-independent parameter is consistent with the maximum manually measured dimension of the dimer structure (~180 Å), and is significantly larger than the maximum dimension of the monomer structure (~124 Å)

Supplementary Fig. 4). Below this lies a network of mainly hydrophobic contacts mediated by the stacking of Trp 82, Ile 74, Leu 69, Thr 114 and Arg 127 (Fig. 1c). A further 10 residues are involved in minor contacts at the D1-D1 interface (Supplementary Figs. 3 and 4). Overall, the D1-D1 dimerization interface is highly conserved since five of the seven key residues at the interface are absolutely conserved across the IgLON family (i.e. Leu 69, Ile 74, Asp 80, Trp 82, and Thr 114), with conservative substitutions at the other positions (Arg 65 is Lys in LSAMP and NEGR1 and Arg 127 is Gln in LSAMP, IGLON5, and NEGR1) (Supplementary Fig. 3).

We also probed the oligomeric state of OPCML in solution using small angle X-ray scattering (SAXS). SAXS has been widely used to give information on size, shape, and orientation of biological macromolecules in solution[6]. The scattering curve calculated from the OPCML dimer fits well with experimental scattering data, while an OPCML monomer does not (Fig. 1e). Pair-distance distribution function analysis[7] (Fig. 1f) showed that WT OPCML has a maximum calculated dimension of 189 Å, in close agreement with the dimer seen in the crystal structure (the monomer has a maximum calculated dimension of 116 Å). These data showed OPCML to be a dimer in solution with a rigid structure.

**Tumor-associated mutations cluster on the OPCML structure.** We mapped the locations of clinical mutations on the crystal structure and analysed their distribution. Ninety eight sites were mutated a total of 243 times (Supplementary Fig. 1B) and, interestingly, some sites were mutated substantially more frequently than average, e.g. P95 was mutated eight times (Supplementary Table 1 and Fig. 1d). When amino acids mutated at higher frequency were mapped on to the crystal structure of OPCML, it appeared that a much higher frequency occurs on one side of the dimer than the other and that the mutations cluster in patches (Fig. 2).

We first focussed on a panel of mutations in D1, where residues at the dimerization interface, glycosylation sites, and exposed residues localized in the aforementioned patches could be investigated for their potential impact on the tumor suppressor properties of OPCML.

**Mutations at the D1 interface disrupt homodimerization.** Members of the IgLON family have been reported to homodimerize and/or heterodimerize for functionality[8,9], but it is not known whether OPCML dimerization is required for its tumor suppressor function. We found that four of the seven salt bridge/stacking residues (Arg 65, Asp 80, Thr 114, Arg 127) and three of the 10 D1-D1 interface residues (Arg 71, Thr 73, Ile 84) are mutated in cancer patients (Supplementary Table 1), suggesting that disruption of dimerization could result in loss of function. To test the role of OPCML dimerization in tumor suppression, we investigated the R65L mutation. Recombinant R65L was secreted

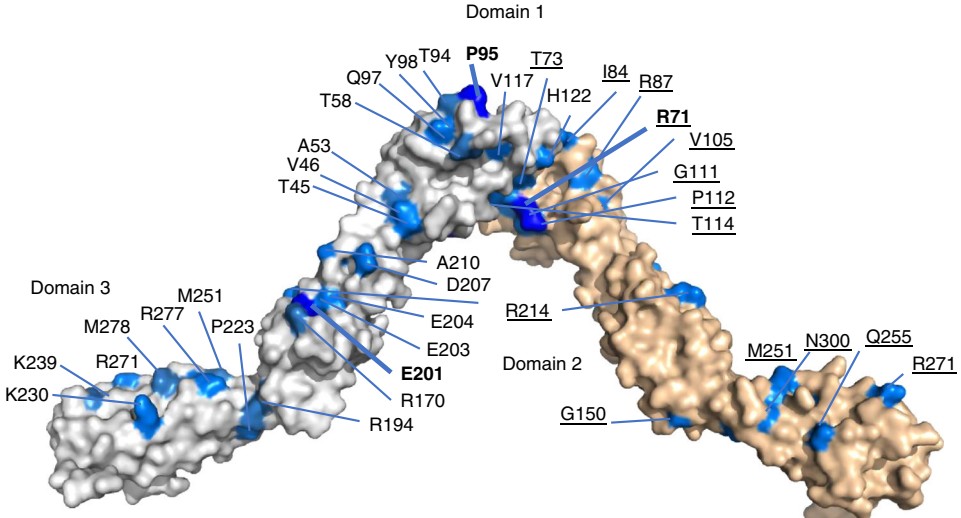

**Fig. 2** Mapping of the most common mutations on to the OPCML crystal structure homodimer. The OPCML homodimer is shown in space-filling representation with one monomer in white and the other wheat. Mutations associated with the wheat monomer are underlined. Mutations in domains 1 and 3 are drawn on the outside of the dimer, whilst domain 2 associated mutations are shown on the inside. The most commonly mutated sites (sites mutated 8 or 9 times) are shown in blue with the residue name in bold. Residues mutated 4, 5 or 6 times are shown in marine and residues mutated 3 times shown in sky blue. For simplicity, mutations found 1 or 2 times are not highlighted, with the exception of K230

at levels similar to WT in insect cells. However, when we examined R65L by size-exclusion chromatography with multiple angle laser light scattering (SEC-MALLS, Supplementary Fig. 6A), or compared it to WT by gel filtration (Supplementary Fig. 6B), we found it to be predominantly a monomer in solution at concentrations up to 32 μM (Supplementary Fig. 6C), with a Kd at least 250-fold weaker than for WT OPCML (Supplementary Fig. 6D). We examined the thermal stability of R65L using a thermal shift assay and found it to melt at 51.5 °C, substantially lower than WT (57 °C, Supplementary Fig. 6E), perhaps due to the loss of quaternary structure. Furthermore, when WT and the R65L mutant were extracted from transduced cells and analysed by SDS-PAGE without prior heat denaturation, WT ran as a dimer while R65L ran exclusively as a monomer (Fig. 3a), demonstrating that WT OPCML exists as a dimer also in cells and that the R65L mutation does indeed disrupt dimerization.

**Loss of glycosylation leads to OPCML aggregation**. We investigated the N70H mutation, which abolishes N-linked glycosylation at this position. Well-ordered glycans were observed at each Asn 70 in the crystal structure, allowing modelling of NAG-NAG-MAN glycosylation sequences. Eleven amino acids make specific contacts with Asn 70 and the associated sugar residues (Supplementary Figs. 3 and 4), and we found five of these to be mutated in cancer patients (Arg 71, Thr 73, Asp 85, Arg 87, and Asp 109). Asn 70 and the first NAG residue are completely buried by D1 and lie close to, but do not make specific contacts with, the reciprocal D1. However, some residues that contact Asn 70 and the glycans are also identified as D1-D1 contact residues (i.e. Arg 71, Thr 73, Leu 75, and Ser 83). Recombinant N70H gave a different SDS-PAGE banding pattern, indicative of altered glycosylation (Supplementary Fig. 6F), and secreted poorly compared to WT OPCML, with expression reduced by at least 80%. Moreover, we observed recombinant N70H to be an unstable protein. It gave a strong fluorescence signal at the beginning of the thermal shift assay (Supplementary Fig. 6E) and exhibited substantial aggregation by gel filtration (Supplementary Fig. 6B). This made analysis by SEC-MALLS impossible. Thus, N70H disrupts OPCML glycosylation and protein stability.

**D1 surface mutations do not affect protein stability**. We previously observed a mutation at Pro 95 in a patient with ovarian cancer[2] and our sequencing analysis revealed several additional mutations at this position (Supplementary Table 1). Pro 95 is located in a type I β–turn, between β-strands D and E of D1 (Fig. 1d), in a solvent-exposed position (starred on Fig. 1a, b) distal from the dimerization interface. Mutations at Pro 95 could lead to disruption of this β–turn and hinder protein folding. However, recombinant expression levels of P95R were indistinguishable to that of WT, and P95R migrated similarly on SDS-PAGE (Supplementary Fig. 6F). We also examined the thermal stability of P95R and found it to melt at 58 °C, similar to WT (57 °C, Supplementary Fig. 6E). Further, P95R was found to be a dimer in solution using SEC-MALLS and migrated similarly to WT by gel filtration (Supplementary Figs. 6A and B). Thus, P95R does not disrupt dimerization or protein stability.

**R65L, N70H and P95R disrupt tumor suppressor function**. To evaluate the effect of these mutations on the tumor suppressor function of OPCML, we used three different ovarian cancer cell lines: SKOV3 and PEA1, which have mesenchymal characteristics, and PEO1, which has epithelial characteristics[10]. These were stably transduced with WT or mutant OPCML (R65L, N70H, P95R) or an empty control vector (CTRL) and characterized. All the corresponding proteins were expressed in the three ovarian cancer cell lines at similar levels as shown by western blot (Fig. 3a), and immunofluorescence microscopy confirmed that they were membrane-localized (Fig. 3b). When tested for their ability to invade or migrate in transwell assays (with or without Matrigel respectively) and for their anchorage-independent growth, WT OPCML completely abolished cell migration in SKOV3 and PEA1 (PEO1 cells did not migrate at all), and invasion and growth in all three cell lines compared to control cells (Fig. 3c–e and Supplementary Fig. 7). However, all the mutants showed loss of function in the three phenotypic assays (Fig. 3c–e and Supplementary Fig. 7).

As we had previously shown that a recombinant form of OPCML protein added exogenously to cancer cells can recapitulate the tumor suppressor effect observed when the protein is expressed endogenously by the cells themselves[5], we

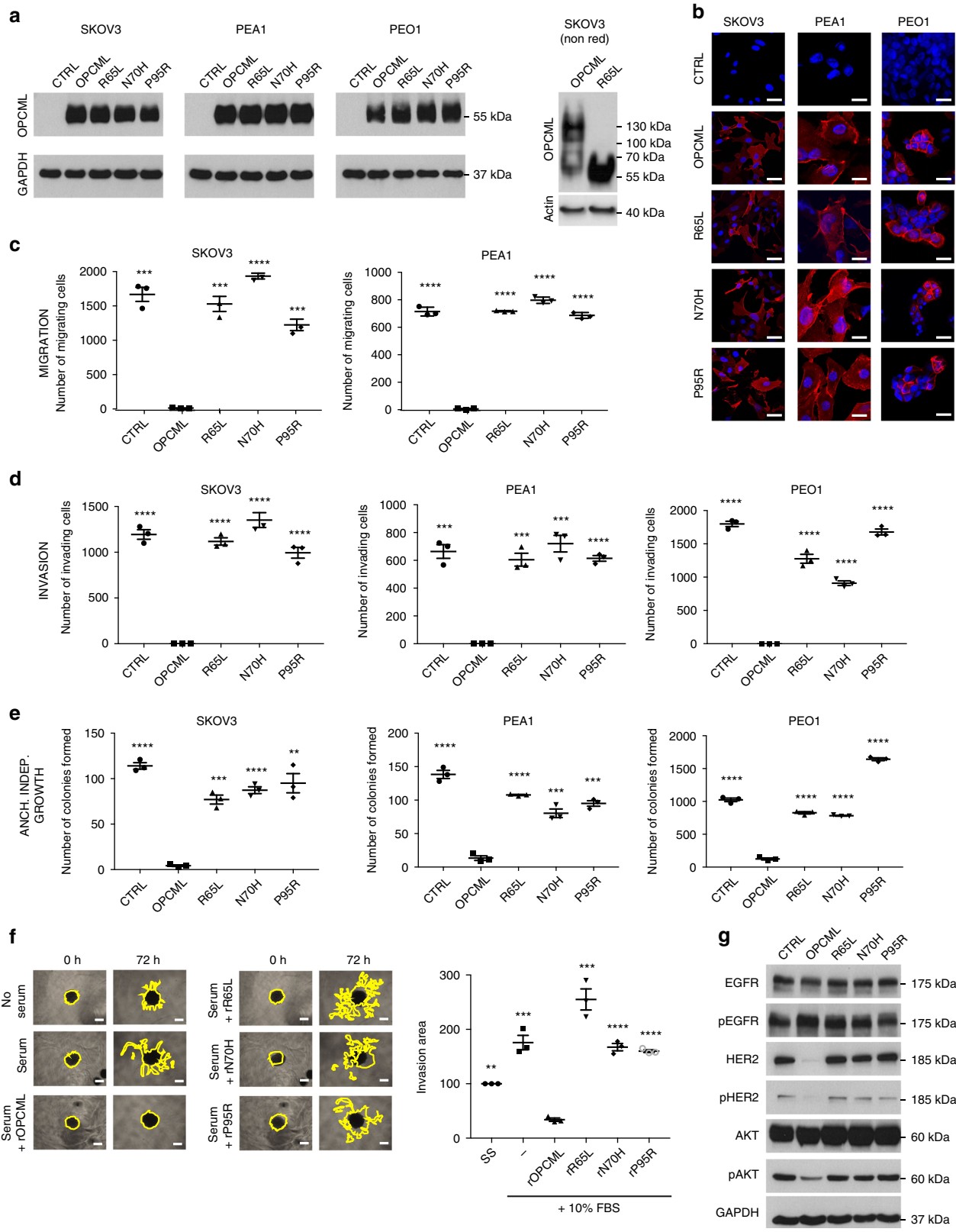

investigated the effect of adding mutant recombinant OPCML proteins in a 3D invasion model. Whereas pre-treatment with WT OPCML protein completely blocked the serum-stimulated invasion of cancer cells into the matrix, the cells pre-treated with mutant protein were unable to impair invasion, corroborating our findings in transduced cells (Fig. 3f).

Furthermore, under serum-stimulated conditions, we observed that the expression of WT OPCML in transduced cells strongly reduced the levels of total HER2 and phospho-HER2, in line with our previous results[5], but it also decreased the levels of phospho-AKT without affecting its total levels (Fig. 3g), while all the mutants had lost this function. Similarly, expression of WT

**Fig. 3** R65L, N70H, and P95R mutants show loss of their tumor suppressor function in vitro. SKOV3, PEA1 and PEO1 ovarian cell lines were transduced with an empty vector (CTRL), wild-type (OPCML) or mutant (R65L, N70H, and P95R) OPCML and analyzed for protein expression by western blot (**a**) and protein expression/localization by confocal microscopy (**b**) with an anti-OPCML antibody. GAPDH or actin were used as loading controls in **a**. Samples in the right panel in **a** were run without heat denaturation. Scale bar = 20 μm. Cells were then tested for migration (**c**) and invasion (**d**) in transwell assays and for anchorage-independent growth in agarose (**e**). All the graphs show the mean ± s.d. of three independent experiments. Student t-test compares CTRL and mutants to wild-type OPCML: *$p < 0.05$, **$p < 0.01$, ***$p < 0.001$, ****$p < 0.0001$. **f** SKOV3 cells grown as spheroids were serum-starved, treated with the indicated recombinant proteins (rOPCML, rR65L, rN70H, rP95R), embedded into Matrigel and then stimulated with serum. Invasion of the spheroids was quantified after 3 days. Scale bar = 250 μm. The graphs show the mean ± s.e.m. of three independent experiments. Student t-test compares controls and mutant recombinant proteins to wild-type rOPCML: **$p < 0.01$, ***$p < 0.001$. **g** SKOV3 cells expressing the indicated wild-type and mutant OPCML were lysed and analyzed by western blot for the expression of total and phosphorylated levels of EGFR, HER2 and AKT. GAPDH was used as loading control

OPCML in the colorectal cancer cell line HCT116 (Supplementary Fig. 8A) decreased the phosphorylation of ERK1/2 but the OPCML mutants could not (Supplementary Fig. 8B).

**D1 mutants have distinctive interactions with RTKs.** To better understand the mechanism behind the loss of function of mutants and highlight possible differences, we analysed their ability to interact with AXL and FGFR1, two known RTK partners of OPCML. As we previously showed[11], the interaction between AXL and OPCML is triggered by the binding of AXL to its ligand GAS6 (Fig. 4a). We observed by Proximity Ligation Assay (PLA, or DuoLink) that this is the case also for the interaction between FGFR1 and OPCML, which is stimulated by the addition of the ligand FGF1 (Fig. 4b). When we analysed the monomeric R65L, we found that it could not bind to either AXL or FGFR1, even upon stimulation with the ligands (Fig. 4a, b). Unexpectedly, the aggregated mutant N70H interacted with both receptors even in the absence of ligands, perhaps due to its compromised folding. Interestingly, we discovered that the surface-exposed P95R mutant could still bind to FGFR1 (Fig. 4b), but it had lost its ability to interact with AXL even after ligand stimulation (Fig. 4a) and this selective loss of binding was confirmed by a mammalian 2-hybrid assay (Fig. 4c, d). Similar results were obtained also in HCT116 cells, where the addition of Gas6 stimulated the binding of AXL to WT OPCML but not to the R65L and P95R mutants (Supplementary Fig. 8C).

In order to further define the functional consequences of these differential interactions, we analysed the effect of the different mutations in a motility assay in SKOV3 cells stimulated with GAS6 or FGF1. Control cells responded to the stimulation and closed the gap, while WT OPCML blocked both GAS6- and FGF1-induced motility (Fig. 4e). As expected from the PLA assay, the R65L mutant, unable to interact with either AXL or FGF1, could not prevent this stimulated motility (Fig. 4e). On the contrary, the N70H mutant, which interacts with both AXL and FGFR1, could inhibit gap closure and showed a phenotype similar to WT OPCML (Fig. 4e). As expected, the P95R mutant could still interact with FGFR1 and prevent FGF1-stimulated motility, but, having lost its interaction with AXL, was unable to block GAS6-mediated migration (Fig. 4e).

We then investigated the effect of the mutations on the inhibition of downstream signalling. Typically, SKOV3 cells respond to GAS6 and FGFR1 stimulation by inducing sustained phosphorylation of the downstream AKT and ERK pathways and OPCML-mediated inactivation of these downstream pathways becomes visible 3 h after stimulation (Fig. 4f, g). As expected, the monomeric R65L mutant could not inhibit the sustained activation of AKT and ERK, while the N70H mutant inhibited the initial activation, delaying the phosphorylation of both AKT and ERK (Fig. 4f, g). In accord with the previous results, the P95R mutant could not impair GAS6-induced signalling, but it completely hindered FGF1 signalling (Fig. 4f, g). Additionally,

when we tested other clinically relevant amino acid substitutions at P95 (P95L and P95S) in SKOV3 cells (Fig. 5a, b) in the context of AXL inhibition, we found that these mutants did not interact with AXL (Fig. 5c) and had no inhibitory effect on GAS6-stimulated signalling (Fig. 5d), migration (Fig. 5e) or invasion (Fig. 5f), underlying the importance of Proline 95 in the interaction of OPCML with AXL.

As we previously found that expression of OPCML confers sensitivity to the AXL inhibitor R428[11], we tested the R65L, N70H and P95R mutants in this context too. They all showed IC50 values and caspase 3/7 activation levels very similar to those of control cells (Fig. 4h, i), confirming that all three mutations abolish the ability of OPCML to permanently suppress AXL.

When we looked at the involvement of the single D1/D2/D3 domains in the OPCML-AXL interaction upon expression in SKOV3 cells (Supplementary Fig. 9A) we found that D1 had a stronger interaction with AXL in full medium compared to D2 and D3 (Supplementary Fig. 9B) and it also responded to stimulation with Gas6 (Supplementary Fig. 9C). These results reinforce the importance of this distal OPCML domain for the regulation of AXL.

**Arg71, the most frequently mutated amino acid in D1.** Arg71 is the most commonly altered residue in D1 (Supplementary Table 1) being mutated nine times: seven times to cysteine and once to histidine in colorectal cancer and glioblastoma. Arg71 is located on the underside of the D1-D1 interface, and makes contacts with the sugar moiety attached to asparagine 70 (Fig. 2 and Supplementary Fig. 4). We expressed R71C in SKOV3 cells (Fig. 6a, b) and analysed its function in phenotypic assays. We observed that R71C had lost its ability to inhibit migration (Fig. 6c), invasion (Fig. 6d) and anchorage-independent growth (Fig. 6e). Furthermore, it was unable to block ERK1/2 activation in HCT116 colorectal cancer cells (Supplementary Fig. 8B).

**Mutations in domains 2 and 3.** Most D2 mutations (39 of the 69) are located between E201 and R214 (Supplementary Fig. 10) and 24 of these mutations involve 4 surface-exposed residues found close together on the D2 surface (R170, E201, S203, and R214) (Fig. 2), Conversely, in D3 the location and incidence of mutations seem to be spread evenly across the length of this domain (Supplementary Fig. 10), most of the high frequency mutations are localized in a stretch on the top portion on D3 (Fig. 2). We selected three mutations from D2 (E201Q, S203R, and R214Q) and three mutations from D3 (K230R, K239N, M278I) for further study. These six mutations were stably expressed in SKOV3 cells (Fig. 6a, b) and their functions analysed. Whilst the expression of WT OPCML impaired cell migration (Fig. 6c) and invasion (Fig. 6d) in transwell assays upon serum stimulation, all the D2 and D3 mutants lost these functions (Fig. 6c, d), as seen for the D1 mutants. Similarly, the

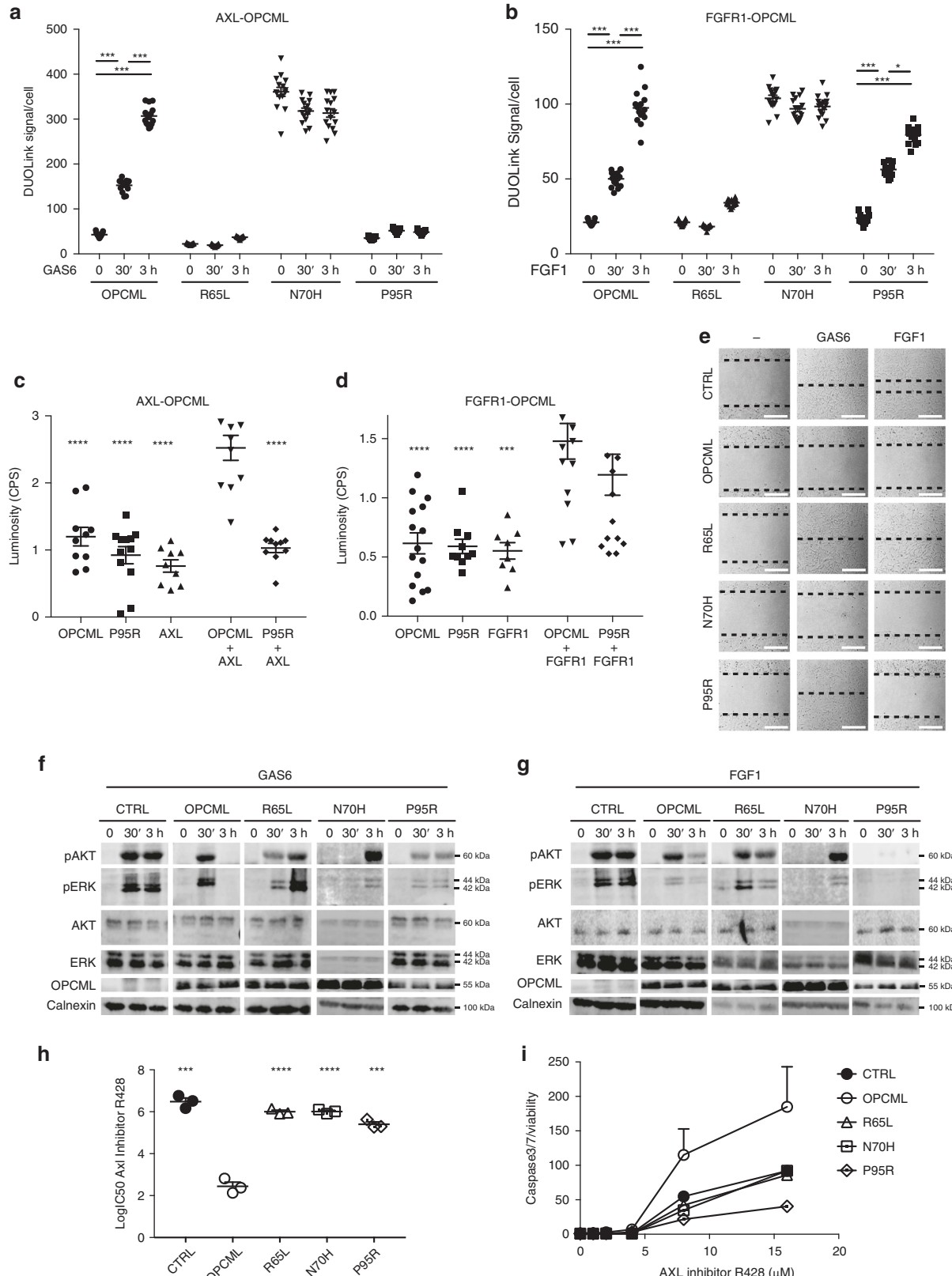

E201Q and the M278I mutants showed loss of function in HCT116 cells (Supplementary Fig. 8B). However, when we tested anchorage-independent growth in SKOV3 cells, all OPCML clinical mutations demonstrated loss of function except for S203R (Fig. 6e) suggesting different roles for the different mutations.

**OPCML inhibits proliferation via its domains 2 and 3.** Since OPCML re-expression strongly decreases growth rate of cancer cells[2], the whole panel of mutants in the three domains was tested in proliferation assays. Whereas D1 mutants did not lose the ability to inhibit proliferation, all mutants in D2 and D3 were

**Fig. 4** R65L, N70H, and P95R mutants interact differently with AXL and FGFR1. SKOV3 expressing an empty vector (CTRL), wild-type (OPCML) or mutant (R65L, N70H and P95R) OPCML were simulated for 30 min or 3 h with GAS6 (**a**) or FGF1 (**b**), and the interaction between OPCML and AXL (**a**) or OPCML and FGFR1 (**b**) was analyzed and quantified by Proximity Ligation Assay (DuoLink). The values from three independent experiments with five images each are shown all together. The bars indicate the mean ± s.e.m. Student t-test: *$p < 0.05$, ***$p < 0.001$. **c, d** Mammalian 2-hybrid assay. Cells were transfected with the indicated single constructs and the relative empty plasmids (not indicated), or with both plasmids expressing OPCML or P95R plus AXL (**c**) or plus FGFR1 (**d**). The bars indicate the mean ± s.e.m. calculated from the values of all the repeats from three independent experiments. Student t-test compared to wild-type OPCML plus AXL (**c**) or OPCML plus FGFR1 (**d**): ***$p < 0.001$, ****$p < 0.0001$. **e** Gap closure assay. SKOV3 cells expressing the indicated wild-type and mutant OPCML were serum starved and then left without stimulation (−) or stimulated with GAS6 or FGF1 as indicated. Migration was imaged after 15 h, and the migration front is highlighted by the dotted lines. Scale bar = 200 µm. **f, g** SKOV3 cells expressing the indicated wild-type and mutant OPCML were serum-starved and then stimulated with GAS6 (**f**) or FGF1 (**g**) for 30 min or 3 h. Cell lysates were analyzed by western blotting for the expression of total and phosphorylated AKT and ERK. Calnexin was used as loading control. **h, i** SKOV3 cells expressing the indicated wild-type and mutant OPCML were treated with the AXL inhibitor R428 for 2 days to assess the IC50 (**h**) or for 1 day to measure apoptosis (**i**). Student t-test compares CTRL and mutants to wild-type OPCML: ***$p < 0.001$, ****$p < 0.0001$. Both graphs show the mean ± s.e.m. of three independent experiments

unable to inhibit growth (Fig. 7a), suggesting loss of function through functional domain specificity.

**OPCML promotes binding to collagen I mainly via domain 1.** Since the composition of the extracellular matrix is known to have a key role in ovarian cancer progression and dissemination[12], we investigated whether OPCML could promote or inhibit adhesion on a variety of extracellular matrix components. We found that WT OPCML promotes cell adhesion on collagen I (Fig. 7b) and collagen IV (Fig. 7c), and that it hinders adhesion on fibronectin (Fig. 7d), laminin (Fig. 7e) and fibrinogen (Supplementary Fig. 11). We then analysed the adhesion properties of the mutants and found that point mutations in each of the domains were sufficient to restore binding to fibronectin, laminin and fibrinogen (Figs. 7d, e and Supplementary Fig. 11). However, only mutations in D1, and to a lesser extent in D2, showed a strongly attenuated adhesion to collagen I (and partially to collagen IV) (Fig. 7b, c).

**OPCML mutants block tumor suppressor functions in vivo.** The results presented above demonstrate the strong impact that single point mutations have in inactivating OPCML's tumor suppressor function(s) in vitro. To expand these observations, we tested the ability of these mutants to block tumorigenicity in vivo. We examined first the R65L, N70H, and P95R D1 mutants, using SKOV3 ovarian carcinoma cells injected intraperitoneally in immunocompromised athymic mice. Cells expressing WT OPCML did not form visible tumors nor ascites in any of the injected mice, while cells expressing the vector control and the three mutants were able to give rise to tumor dissemination and ascites, though to different extents depending on the mutation (Fig. 8a). In particular, the R65L and P95R mutants formed tumors and ascites very similarly to control cells (Fig. 8a), while the N70H mutant, which is aggregated and had a more mixed phenotype in vitro, presented a more attenuated phenotype and it gave rise to small tumors in some mice (Fig. 8a). We also tested representative mutations in D1 (R65L), D2 (R214Q), and D3 (M278I) in the chick chorio–allantoic membrane (CAM) assay (Fig. 8b). The R65L mutant was chosen also as a measure to compare the results in mice and CAM models. As evident in Fig. 8b, the expression of WT OPCML prevented tumor growth on the CAM, while all three clinical mutations were impaired in this function and the cancer cells expressing them grew at least as well as the control cells.

## Discussion

OPCML is a frequently inactivated tumor suppressor in multiple types of human cancer, predominantly through somatic methylation of the promoter region[1,2,13–25]. We found 287 somatic

mutations in the *OPCML* gene from sequencing data within the COSMIC and TCGA databases. Though these mutations remain rare, the same amino acid was found mutated in patients with different cancer types, highlighting the relevance of those residues. For this reason, we solved the crystal structure of OPCML, the first reported structure from the IgLON family, and mapped the clinical mutations to its 3D structure in order to possibly help explain the multiple phenotypes of this tumor suppressor protein.

We previously reported that OPCML interacts with a subset of RTKs and induces their degradation and/or dephosphorylation when re-expressed in ovarian cancer cells[5,11]. This down-modulation of the RTK network inhibits signalling and phenotypic assays defining hallmarks of cancer both in vitro and in vivo[2]. However, it was unknown which regions of OPCML were implicated in the various phenotypes or whether the clinical mutations identified here caused loss of OPCML function.

OPCML is composed of three canonical Ig-like domains with a C-terminal GPI anchor localized in the plasma membrane outer leaflet. Here we demonstrate that OPCML forms a homodimer in the asymmetric unit, mediated by extensive interactions among residues in the N-terminal domain D1. We confirmed the homodimeric arrangement in solution using SAXS, SEC-MALLS, and gel filtration. Based on the high degree of amino acid conservation at this interface in the other IgLON members, we predict that the same quaternary structure will be found throughout the IgLON family. With a large proportion of mutations being present in dimerization domain D1, we hypothesized an important role for D1 in mediating OPCML's tumor suppressor functions. To test this hypothesis, we characterized a protein with the R65L clinical mutation and found this to be predominantly monomeric in solution, as evidenced by SEC-MALLS and gel filtration, and also in cells as demonstrated by non-heated SDS-PAGE. Indeed, this mutant, unlike WT and despite it being correctly expressed in cells, was unable to block migration, invasion and anchorage-independent growth, or in vivo tumorigenicity because it could not interact with RTKs like AXL and FGFR1 and was thus incapable of inhibiting their signalling. We noted that OPCML's affinity for RTKs is strongly increased upon their RTK dimerization/activation following ligand binding. Thus it is likely that dimerization of OPCML is necessary for RTK dimer interactions, thus explaining the observation of the functional defect in monomeric R65L.

We evaluated N70H clinical mutation to test the effect of mutating a site that is both glycosylated and buried deep within the D1-D1 interface, we found the resultant recombinant protein (secreted using insect cells) to be expressed at low levels and to be unstable. Although N70H could still bind to RTKs even without stimulation, which was probably due to its aggregated nature, it demonstrated a loss of function phenotype in most of the in vitro assays and an intermediate phenotype in vivo.

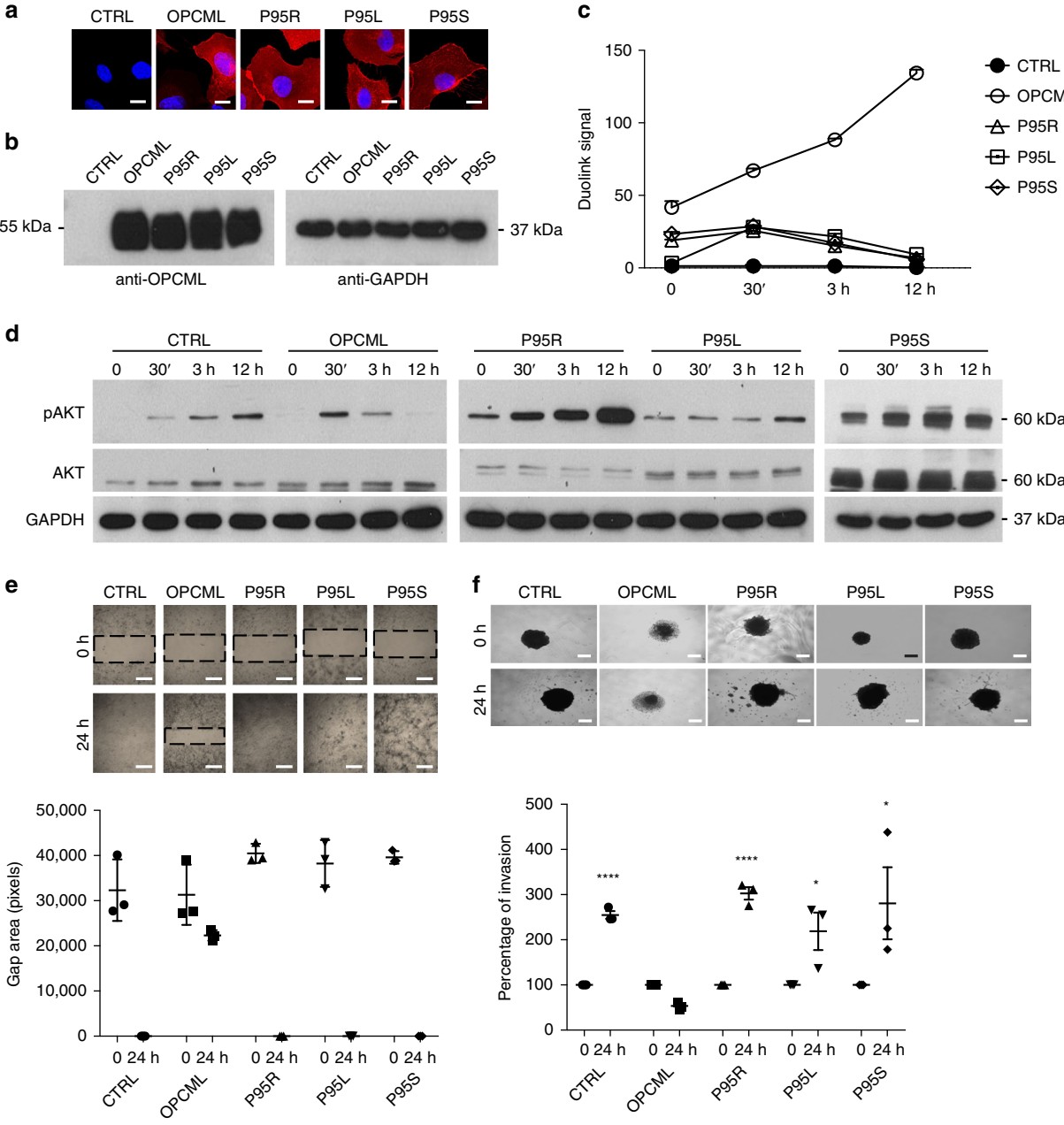

**Fig. 5** Mutations at P95 show similar loss of tumor suppressor function and AXL interaction. SKOV3 cells were transduced with an empty vector (CTRL), wild-type (OPCML) or mutant (P95R, P95L, P95S) OPCML and analyzed for protein expression/localization by confocal microscopy (**a**) and protein expression by western blot (**b**) with an anti-OPCML antibody. Scale bar = 5 μm. Cells were starved and then stimulated with GAS6 for 30 min, 3 h or 12 h as indicated, and the interaction between OPCML and AXL measured by PLA (**c**), the signaling to pAKT detected by western blotting (**d**), the migration tested by gap closure assay (**e**) and the invasion studied by 3D spheroid invasion assay in Matrigel (**f**). GAPDH was used as loading control in **b**, **d**. The images were taken with a confocal microscope in **a** or an inverted microscope in **e** and **f**. Scale bar = 200 μm (**e**) and 250 μm (**f**). All the graphs show the mean ± s.e.m. of three independent experiments. Student *t*-test compares CTRL and mutants to wild-type OPCML at 24 h of invasion: *$p < 0.05$, ****$p < 0.0001$

The crystal structure of OPCML presented here came from insect cell-expressed material but much of the functional data employed OPCML from a mammalian source. It is possible that interactions between glycan at asparagine 70 and protein at the D1-D1 dimer interface could be different depending on the origin of the material. Insect cell and human N-linked glycans typically differ at terminal sialylation sites (present in human not insect) and occasionally by addition of α1,3-linked fucose at the initial GlcNac[26]. We did not observe ordered density extending the terminal glycans for any of the glycans modeled, nor did we observe α1,3-linked fucose at the initial GlcNac. Of note, when

the recombinant proteins produced in insect cells were tested in phenotypic assays (spheroid invasion, Fig. 3f), they showed a loss of function similar to the mutant proteins produced directly by the cells. These data indicate functional equivalence between insect- and mammalian-produced OPCML proteins.

Proline 95 was previously found to be mutated to arginine causing loss of tumor suppressor phenotype[2] and from our sequence analysis we found it to be mutated eight times to three different amino acids (the third most commonly mutated residue after R71 and E201) in patients. The P95R mutation is distal from the dimerization interface, and the resultant protein was

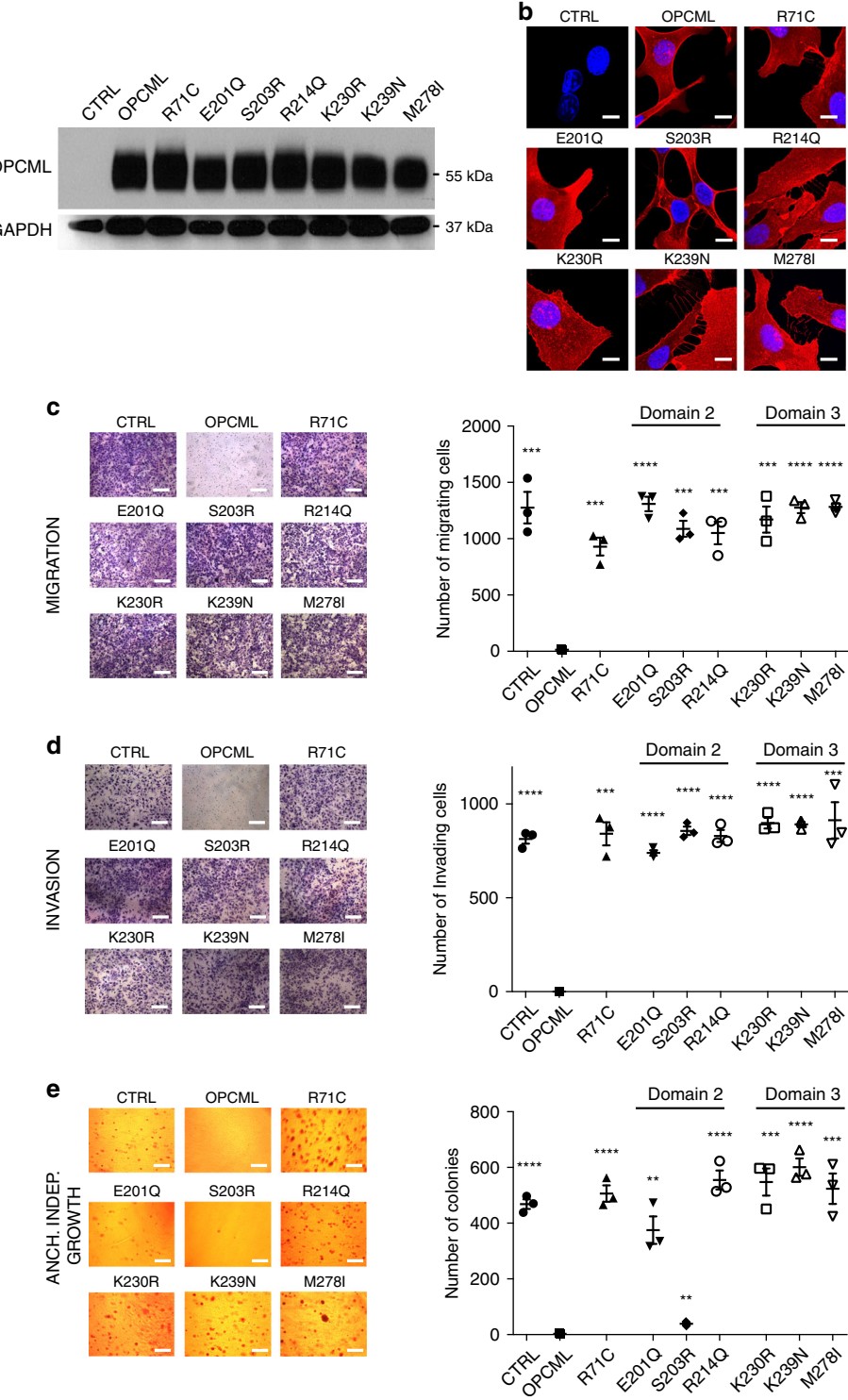

**Fig. 6** Point mutations in D2 and D3 lead to loss of function. SKOV3 cells were transduced with an empty vector (CTRL), wild-type (OPCML) or mutant (R71C, E201Q, S203R, R214Q, K230R, K239N, M278I) OPCML and analyzed for protein expression by western blot (**a**) and protein expression/localization by confocal microscopy (**b**) with an anti-OPCML antibody. GAPDH was used as loading control in **a**. Scale bar = 5 μm. Cells were then tested for migration (**c**) and invasion (**d**) in transwell assays and for anchorage-independent growth in agarose (**e**). Scale bar = 50 μm (**c**, **d**) and 200 μm (**e**). All the graphs show the mean ± s.e.m. of three independent experiments. Student *t*-test compares CTRL and mutants to wild-type OPCML: **$p < 0.01$, ***$p < 0.001$, ****$p < 0.0001$

homodimeric in solution, as shown by SEC-MALLS and gel fil-tration. In addition, this protein reached the plasma membrane in cells, and the secreted recombinant protein had a melting temperature similar to WT, indicating that the loss of tumor

suppressor phenotype was not due to the stability of the molecule.

Despite the different chemical nature of the substitutions (to Ser, Arg, or Leu), the disruptive effect of all three mutations was

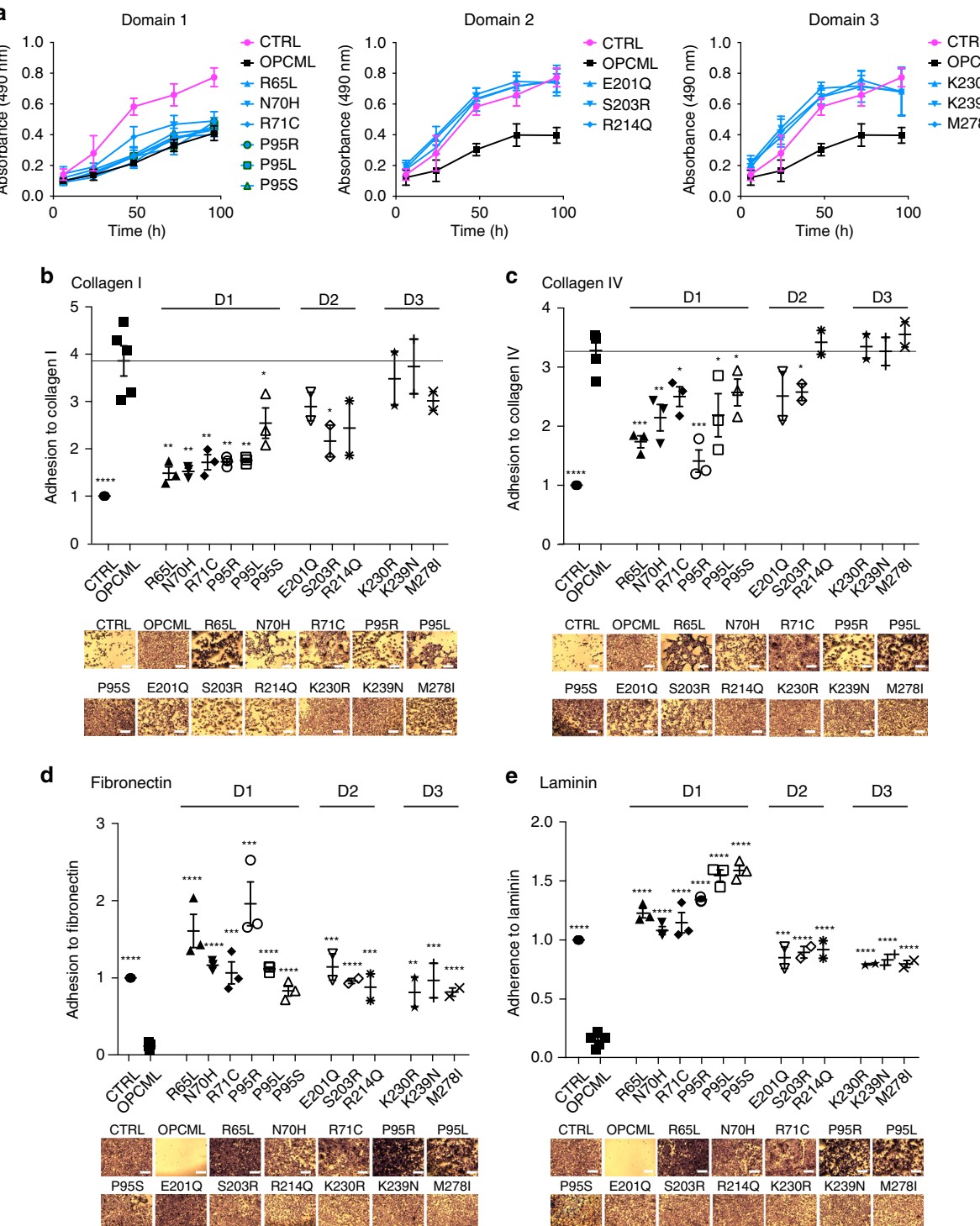

**Fig. 7** Different domains mediate distinct functions. **a** SKOV3 cells transduced with an empty vector (CTRL), wild-type (OPCML) or mutant (R65L, N70H, R71C, P95R, P95L, P95S in D1 in the left panel, E201Q, S203R, R214Q in D2 in the central panel, K230R, K239N, M278I in D3 in the right panel) OPCML were analyzed for proliferation. CTRL cells are represented in magenta, wild-type OPCML in black and the mutants in light blue. All the graphs show the mean ± s.d. of four independent experiments. **b–e** The different cell lines were plated onto collagen I (**b**), collagen IV (**c**), fibronectin (**d**) or laminin (**e**), and adhesion was tested after 1 h. Values have been normalized to CTRL. Scale bar = 50 μm. All the graphs show the mean ± s.e.m. of three independent experiments for domains 1 and 2 independent experiments for domains 2 and 3. Student t-test compares CTRL and mutants to wild-type OPCML: *p < 0.05, **p < 0.01, ***p < 0.001, ****p < 0.0001

similar in our assays and intriguingly, we found that Pro95 mediates the interaction with AXL but not with FGFR1. The fact that mutations at P95 had selectively lost the interaction with AXL, the downstream signalling response to GAS6 and the

cognate functional in vitro phenotypes, demonstrates the structural and functional basis of this clinical mutation. Furthermore, the fact that in vivo suppression of tumorigenicity was lost with P95R indicated the biological importance of the

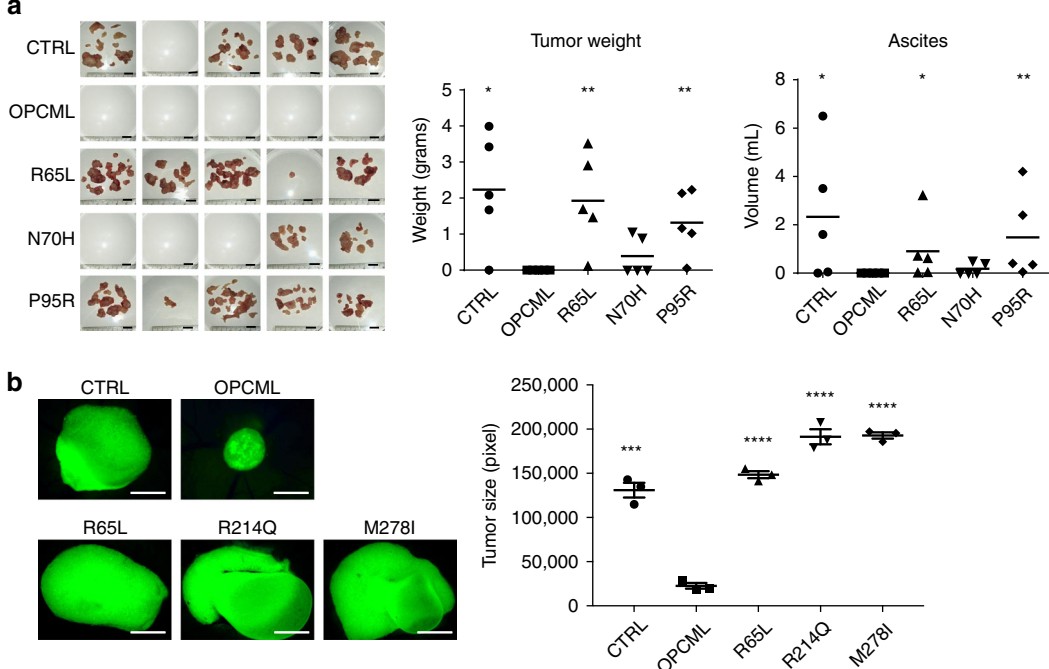

**Fig. 8** Point mutations impair OPCML tumor suppressor function in in vivo models. **a** SKOV3 cells expressing empty vector (CTRL), wild-type (OPCML) or mutant (R65L, N70H and P95R) OPCML were injected intraperitoneally in five female athymic mice each. Tumors and ascites were collected and measured in grams and milliliters, respectively, as indicated. The mean is indicated by a line in the dot plots. Student *t*-test compares CTRL and mutants to wild-type OPCML: *$p < 0.05$, **$p < 0.01$. Scale bar = 1 cm. **b** SKOV3 cells expressing GFP and the indicated constructs were grown embedded in Matrigel on top of chick chorio–allantoic membranes. Tumors were excised, imaged and the GFP-positive area measured. Scale bar = 500 μm. The graph shows the mean ± s.e.m. of three eggs. Student t-test compares CTRL and mutants to wild-type OPCML: ***$p < 0.001$, ****$p < 0.0001$

AXL pathway in ovarian cancer despite an intact FGFR1/FGF1 axis.

Additionally, the results described here link IgLON proteins to extracellular matrix adhesion, demonstrating that OPCML promotes adhesion to collagen but inhibits adhesion to other substrates such as fibronectin and laminin. The mechanism of this selectivity remains to be established, as OPCML could interact directly with these substrates and/or interfere with the activity of other adaptor molecules that bind directly to them. It is interesting to note that residues in each of the three domains are required to inhibit binding to fibronectin and laminin, while adhesion to collagen I and IV is D1-specific, clearly pointing to the existence of diverse mechanisms for promotion or inhibition of adhesion to extracellular matrix components.

The only phenotype that did not demonstrate a dependency on D1 residues was proliferation in vitro, with D1 mutations continuing to inhibit proliferation, whereas mutations in D2 and D3 lost the ability to suppress proliferation. This suggests that D2 and D3 could mediate the interaction with the binding partners responsible for inhibition of proliferation.

It is also curious to note that out of the 12 mutants tested, the only mutation that did not completely impair OPCML's ability to inhibit anchorage-independent growth was S203R. Although this mutation is very close to the E201Q substitution, S203R and E201Q give opposite phenotypes in this particular assay. It would be interesting to identify the differential binding partners of these two mutants to dissect this specific suppressive function of OPCML.

Concerning tumorigenicity, we had previously shown[2] and we confirm here that OPCML is a potent tumor suppressor that abrogates tumor growth in in vivo mouse models. Here we demonstrate that dimerization is key to this function (via the R65L mutant), and also that the interaction with specific binding partners (P95R) is required to inhibit tumorigenesis. The CAM

assay gave similar results and it also demonstrated that mutations in D2 and D3 affect the inhibition of tumorigenicity, pointing to the importance of the integrity of the three domains for OPCML's full tumor suppressor activity in vivo at least in ovarian cancer.

The quaternary structure of the OPCML homodimer presented here looks highly similar topologically to that of nectins[27]. Nectins are composed of three consecutive Ig-like domains, are also plasma-membrane-associated and can mediate cell-cell communication via pairwise cis- and trans- interaction homo- or heterotypically with other nectin family members. The IgLON family of proteins have been suggested to pair in a similar fashion[8,9] and for these reasons a possible common mechanism could be suggested.

In summary, in this work we present the crystal structure of OPCML, identify new phenotypes of this potent tumor suppressor and map diverse functions to distinct protein structural domains through an understanding of structure/function relationships associated with clinically occurring cancer missense point mutations that impair OPCML's function. Future work will help to further dissect the mechanisms behind these phenotypes, with the aim of developing OPCML as a potential therapeutic[28] in cancer patients.

## Methods

**Protein expression and purification**. WT human OPCML (UniProt entry Q14982), R65L, N70H, and P95R were expressed as Fc-fusion proteins in insect cells after baculovirus infection using Hi5 cells grown in suspension in SFX-insect serum-free medium. Specifically, OPCML residues 36–316, corresponding to D1-D3 were cloned in to the pFastBac1 vector (Invitrogen, USA) downstream of an N-terminal ceruloplasmin secretion signal (MKILILGIFLFLCSTPLQ) and upstream of a C-terminal portion composed of ALVPRGS and human IgG1 (Uniprot entry P01857, residues 118–330). The insect cell supernatant was harvested 4 days post infection and applied to protein A affinity resin (Repligen). The resin was washed extensively with PBS pH 7.4 and the OPCML-IgG1 fusion eluted in 50 mM glycine pH 3.0. After neutralization and buffer exchange into 20 mM Tris-HCl pH 7.5, 100 mM NaCl and 1 mM CaCl₂, the IgG1 tag was removed by

thrombin digestion. Cleaved OPCML was then purified to homogeneity by negative purification on protein A resin, HiTrap Q HP anion exchange chromatography (GE Healthcare) and finally size-exclusion chromatography.

**Crystal structure determination**. For crystallization, WT OPCML was concentrated to 10 mg ml$^{-1}$ in 20 mM Tris-HCl pH 7.5, 100 mM NaCl, and 0.01% NaN$_3$. Rectangular pyramid-shaped crystals measuring $300 \times 300 \times 400$ μm were obtained at room temperature by mixing 1 μL of OPCML with 1 μL of precipitant solution. The best precipitant was found by an iterative process of mixing initial crystal screen hits back in to the screens[29] and was a 50/50 mix (v/v) of 1.6 M MgSO$_4$, 100 mM MES pH 6.5 and 20% PEG 8000, 100 mM HEPES pH 7.5 using the sitting drop vapor diffusion technique. Platinum derivatives were created by soaking OPCML crystals overnight in mother liquor supplemented with 10 mM K$_2$PtCl$_4$. Crystals were flash-frozen by mixing 50% mother liquor (v/v) with 50% saturated Li$_2$SO$_4$. Around 5% of all crystals diffracted maximally to between 3–2.65 Å resolution.

**Structure determination/phasing**. X-ray diffraction data from native protein crystals ($\lambda = 0.97931$ Å) and Pt-modified crystals ($\lambda = 1.06929$ Å) were collected at 100 °K on the LRL-CAT beamline at the Advanced Photon Source, Chicago, U.S.A. SBGrid collaborative software library was used throughout the structure determination process. Diffraction images were processed using IMOSFLM[30] and scaled using SCALA[31]. Despite extensive searches using a variety of different Ig-domain containing search models, no molecular replacement solution could be found using native OPCML diffraction data. These data exhibited strong Patterson peaks consistent with translational non-crystallographic symmetry that may have impeded conventional molecular replacement. We evaluated several potential heavy-atom derivatives, and obtained useful diffraction data in a different space group from a platinum-soaked crystal. A combined molecular replacement/single-wavelength anomalous dispersion approach (MR-SAD) as implemented in Phenix Autosol[32] provided a strong solution using the isolated Ig domain 49–124 from PDB: 1Z9M. This gave 9 platinum sites per asymmetric unit, a figure of merit of 0.31 and average anomalous difference of 0.0966. Hundred and thirty-five residues including 58 side chains were autobuilt, comprising much of D1 and parts of D2 and D3. Phase improvement by density modification provided electron density sufficient to build 104 additional residues from D2 and D3 using COOT[33] with repetitive rounds of refinement using Phenix Refine[34].

The partial model derived from MR-SAD phasing of the I4$_1$22 Pt derivative dataset was used as a molecular replacement model to provide initial phasing for the somewhat higher resolution P4$_3$2$_1$2 native dataset (Table 1). A strong solution (LLG = 4360 and TFZ = 53.5) was found with two molecules in the asymmetric unit and confirmed by examination of composite omit maps (Supplemental Fig. 12). The two molecules were related by a two-fold rotation axis nearly parallel to the crystallographic (c*) axis, resulting in nearly perfect translational non-crystallographic symmetry and (x+ 1/2, y + 1/2, z + 1/2) pseudocentering. Examination of the diffraction pattern revealed weaker intensity for the h + k + l = 2n spots as compared to the h + k + l (not equal to) 2n, with the h + k + l = 2n spots corresponding to those observed for the Pt derivative in the I4$_1$22 space group. An essentially complete model for the OPCML dimer was built by repeated rounds of model building and refinement as described above. The final PHENIX refinement strategy employed XYZ coordinate, Group-B-factor, TLS and occupancy refinement with secondary structure restraints. Noncrystallographic symmetry restraints (NCS) were applied throughout. This gave a final model with Rwork and Rfree of 28.7/30.4, respectively. Final omit electron density is shown in Supplementary Figure 12.

The overall geometry in the final structure is good, with 96% of residues in favored regions of the Ramachandran plot and no outliers. Clear electron density was visible for residues corresponding to D1-D3 (residues 42–316) apart from a loop between the C and C′ strands in D2 (residues 175–179 in molecule A and 174–178 in B). Structure figures were generated using the program PyMOL[35].

An analysis of OPCML surfaces, the dimerization interface and crystal contacts was made possible due to calculations performed by PISA[36].

**SEC-MALLS**. An estimation of the molecular weight of WT, R65L, N70H and P95R OPCML in solution was given by SEC-MALLS. This was performed on a chromatography system using a refractive index detector (Optilab T-rEX, Wyatt Technology) connected to a multi-angle laser light scattering detector (DAWN HELEOS II, Wyatt Technology). Protein samples (at 1 μg μL$^{-1}$) were injected and ran at 1 mL min$^{-1}$ in PBS (pH 7.4) buffer at room temperature.

**Size-exclusion chromatography and Kd determination**. To determine the binding affinity of OPCML dimerization, various concentrations of WT and R65L OPCML were subjected to size-exclusion chromatography. Briefly, dilutions of purified soluble WT and R65L OPCML were prepared and allowed to equilibrate for 24 h. Fifty microliters of each dilution were run over a Superdex 200 10/30 GL column at 0.5 mL min$^{-1}$. The resulting chromatograms were normalized to account for small baseline shifts and slight differences in injection volume. Peak elution volumes at each dilution were plotted against concentration and fit to the equation EV$_{obs}$ = EV$_{monomer}$ + (1-fractMonomer)(EV$_{dimer}$ − EV$_{monomer}$), where EV$_{obs}$, is the elution volume observed at particular total protein concentration P$_{tot}$,

EV$_{monomer}$, and EV$_{dimer}$ are fit values for monomer and dimer elution volumes, respectively, and fractMonomer is fraction of the total protein concentration represented by the monomer, determined using the formula fractMonomer = $(-K_d + sqrt(K_d^2 + 8*P_{tot}*Kd))/4P_{tot}$, where $K_d$ is the fit value for the dimerization constant[37]. For estimation of a lower bound on the R65L OPML Kd, EVdim was constrained to the same value as for WT OPCML. GraphPad Prism 7 software (USA) was used for curve fitting.

**Thermal stability assay**. Five microgram of purified baculovirus-derived recombinant WT, R65L, N70H and P95R OPCML were incubated in the presence of 10x SYPRO Orange (Sigma-Aldrich, Cat. No. S5692) for 1 min at 25 °C, in PBS buffer (pH 7.4). The fluorescence measurements were recorded in a real-time PCR machine (Bio-Rad CFX96) from 25 °C in 0.5 °C increments until 95 °C.

**Small angle X-ray scattering**. SAXS was measured for a sample of WT OPCML. The sample was dialyzed into 50 mM HEPES pH 7.5, 200 mM sodium chloride, 2% glycerol, 0.02% sodium azide by equilibration in Hampton Research dialysis buttons using a 7 kDa MWCO membrane. Sample was filtered through 0.22 μm Spin-x column to remove potential aggregates and two two-fold dilutions were prepared using dialysate. X-ray scattering was measured using five one-second exposures incident on static sample.

Scattering curves were merged and buffer was subtracted by manual minimization of the water scattering peak at $q = 2$ Å$^{-1}$. Scattering curves for the three concentrations were further merged and the curves were trimmed to the range $0.0080 < q < 0.190$ using SCÅTTER v1.0 software[38].

SAXS data was further analyzed using the Fast SAXS Profile Computation with Debye Formula (FOXS) web server[39]. Experimental data for OPCML was fit to calculated SAXS curves originating from the OPCML crystal structure dimer or monomer model consisting of chain A.

**Cell culture**. All ovarian cancer cell lines were maintained in RPMI-1640 media (Sigma-Aldrich) supplemented with 10% Foetal calf serum (FCS, First link), 0.2 mM L-Glutamine (Gibco), penicillin 50 U ml$^{-1}$, and streptomycin 50 μg ml$^{-1}$ (Gibco). HEK293T (Human Embryonic Kidney) cells were maintained in Dulbecco's Modified Eagle Medium (DMEM) supplemented as above. HCT116 were grown in McCoy's medium supplemented as above. All cell lines were cultured at 37 °C in a 5% CO$_2$ humidified incubator. Transduced cells were maintained in full medium supplemented with Puromycin dihydrochloride (Gibco) (10 μg ml$^{-1}$ for SKOV3, 5 μg ml$^{-1}$ for PEA1, and 1 μg ml$^{-1}$ for PEO1 and HCT116).

SKOV3, HEK293T and HCT116 cells were purchased from ATCC. PEA1 and PEO1 cells were acquired from Cancer Research UK. All cells were tested regularly to exclude mycoplasma infection.

**Site-directed mutagenesis**. Primers were designed using the Quick-change Agilent primer design program available online at www.agilent.com/genomic/qcpd. The primers were designed to have a complementary sequence ranging from 12–20 bases on either side of the desired mutation site. R65L Forward: 5′- GATGACC GGGTAACCCTGGTGGCCTGGCTAAAC-3′; N70H Forward: 5′- CGGGTGGC CTGGCTACACCGCAGCACCATCCTC-3′; R71C Forward: 5′- GTGGCCTGGC TAAACTGCAGCACCATCCTCTAC-3′; P95R Forward: 5′- ATCATCCTGGTC AATACACGAACCCAGTACAGCATCATG-3′; P95L Forward: 5′-ATCATCCT GGTCAATACACTAACCCAGTACAGCATCATG-3′; P95S Forward: 5′-ATCA TCCTGGTCAATACATCAACCCAGTACAGCATC-3′; E201Q Forward: 5′-CA GTCCGGGGAGTACCAATGCAGCGCGTTGAAC-3′; S203R Forward: 5′-GG GGAGTACGAATGCCGCGCGTTGAACGATGTC-3′; R214Q Forward: 5′-GCT GCGCCCGATGTGCAGAAAGTAAAAATCACT-3′; K230R Forward: 5′-CCCTA TATCTCAAAAGCCAGGAACACTGGTGTTTCAGTC-3′; K239N Forward: 5′-G TTTCAGTCGGTCAGAACGGCATCCTGAGCTGT-3′; M278I Forward: 5′-G AAAACAAAGGCCGCATCTCCACTCTGACTTTC.

Mutagenesis reactions were performed following a Quick-Change® Site-Directed Mutagenesis protocol (Agilent) using Pfu DNA Polymerase (Thermo Scientific). The plasmid DNA template, used to generate the desired mutants, was pENTR4 containing the OPCML WT cDNA. The mutagenesis reaction was prepared to make a final volume of 20 μl that contained the following components: up to 20 μl of autoclaved distilled water; 2 μl of 10x buffer + MgSO$_4$; 0.5 μl of 25 mM dNTPs; 1 μl of each mutagenic primer (100 ng μl$^{-1}$); 20 ng μl$^{-1}$ plasmid DNA template; 1 μl of Pfu DNA Polymerase; (Thermo Scientific, UK) and 3% DMSO. The reaction mixture was run in the MJ Research PTC-200 PCR Thermal Cycler—GMI. After the reaction was completed, it was incubated with 1 μl of DpnI restriction enzyme (New England Biolabs®, UK) at 37 °C for one hour. The digested mutagenesis product was then transformed into competent bacterial cells. The mutations were confirmed by DNA sequencing.

**Single domain cloning**. D1 (Pro39-Ser126), D2 (Arg127-Thr219), and D3 (Val229-Tyr310) constructs were purchased from Life Technologies designed as entry clones. All constructs have a signal sequence for secretion in the N-terminus and an HA tag at the C-terminus before the GPI-anchor attachment site.

**The Gateway® cloning system**. Sequenced WT and mutant pENTR4-OPCML plasmids were used to shuttle the sequence of interest to the Destination vector, pLenti CMV Puro DEST. Reactions were carried out mixing 50 ng of entry vector and 150 ng of destination vector with 1X LR Clonase reaction buffer in a total volume of 10 µl. Reactions were incubated at 25 °C for 1 h, after which 4 µg of proteinase K (Gateway$^{TM}$ LR Clonase II Enzyme Mix, Invitrogen) was added and samples were incubated for further 10 min at 37 °C. Then, 1 µl of reaction was used for bacterial transformation using HB101 competent cells. The plasmids were amplified and purified.

**Transfection**. Human Embryonic Kidney (HEK293T) cells were used to produce lentivirus. The cells were seeded into 60 mm tissue culture dishes at $3 \times 10^6$, 24 h before Lipofectamine 2000™ transfection following the reagent protocol (Invitrogen). Four microgram of pLenti CMV Puro DEST were mixed with 3.5 µg of packaging plasmid (p8.9) and 0.5 µg of envelope (pMDG) in 500 µl of Opti-MEM medium, and incubated for 5 min at room temperature. At the same time 20 µl of Lipofectamine™2000 were diluted in 500 µl of Opti-MEM medium and also incubated for 5 min. After the incubation, the two solutions were mixed and incubated at room temperature for a further 20 min. The transfection complex was added to the cells drop-wise, incubated for six hours after which the medium was changed to full growth medium with antibiotics (Penicillin/Streptomycin). After 48 h, the virus was collected and underwent centrifugation at 1000×g for 4 min, after which it was filtered using 0.20 µm filters. Aliquots were made into 1.5 Eppendorf tubes and frozen at −80 °C.

**Transduction**. Ovarian cancer cell lines (SKOV3, PEA1, and PEO1) were seeded into 24-well plates to suitable density, transduced with different amounts of virus (50–450 µl) and incubated for 24 h before changing the medium and adding the selective antibiotic (Puromycin). Cells were then expanded and protein over-expression tested by western blot.

**Whole cell lysate preparation**. Cultured mammalian cells in six-well plate were lysed in 100 µl of RIPA buffer (25 mM TrisHCl pH 7.6, 150 mM NaCl, 1% NP-40, 1% sodium deoxycholate, 0.1% SDS) supplemented with phosphatase (Merck Millipore) and protease inhibitors (Roche and Sigma-Aldrich). Cells were lysed on ice for 15 min, collected into Eppendorf tubes and subjected to centrifugation at 16000×g for 20 min at 4 °C. After centrifugation, supernatants were transferred to new Eppendorf tubes and used for protein concentration determination using the BCA protein assay kit (Pierce), following manufacturer's instructions.

**Western blot**. Protein samples (5–10 µg) were separated on 8–10% gels by SDS-PAGE and transferred onto a nitrocellulose membrane (Merck, Millipore). The membrane was then blocked and incubated with the desired primary antibodies diluted 1:1000: anti OPCML (goat pAb, #AF2777, R&D Systems), phospho-AKT (Thr308 D25E6 XP® rabbit mAb, #13038, Cell Signaling), AKT (pan 11E7 rabbit mAb, #4685 Cell Signaling), phospho-ERK1 (pT202/pY204) + phospho-ERK2 (pT185/pY187) (mouse mAb, ab50011, Abcam), ERK1/2 (rabbit pAb, ab17942, Abcam) and anti-GAPDH (Cell Signalling), over-night at 4 °C. The membrane was washed, then incubated with the appropriate secondary antibodies (DAKO) diluted 1:5000. After final washings, proteins were detected using Immobilon Western Chemiluminescent HRP Substrate system (Millipore) and GE Healthcare Amersham™ Hyperfilm™ ECL film with a Kodak SRX2000 (Rochester, NY, USA) developer machine.

**Immunofluorescence confocal microscopy**. Coverslips with adherent monolayer cells were rinsed twice in PBS, fixed with ice-cold methanol for 5 min at −20 °C and then washed again. The coverslips were blocked with 3% Bovine serum albumin (BSA) for 30 min at room temperature, and then incubated with the anti-OPCML (mouse mAb, #MAB27771, R&D Systems) antibody diluted 1:100 at room temperature for 1 h in a wet chamber, washed and incubated with a suitable secondary antibody diluted 1:400 at room temperature for 30 min. Coverslips were mounted with a drop of Prolong gold 4′, 6-diamidino-2-phenylindole (DAPI) (Invitrogen) and left to dry in the dark at room temperature overnight before being examined under an SP5 confocal microscope (Leica).

**Cell migration and invasion assays**. Cell migration and invasion assays were performed using 8 µm pore PET Membrane chambers for 24-well plates (Corning, UK). Cells were grown in full medium and then re-suspended in serum-free media when added to chambers with or without Matrigel. $2.5 \times 10^4$ cells/chamber were seeded in duplicate. Cells were allowed to migrate/invade for 24 h, after which the cells that had not migrated/invaded were removed with a cotton swab from inside the chamber and the inserts washed in PBS. Cells were fixed with ice-cold methanol and stained with 0.1% crystal violet in 25% methanol. At least six images were taken for each insert at ×10 magnification using a Nikon Eclipse TE2000-U microscope and QCapture Pro software. Analysis and quantitation were performed using ImageJ. The data were obtained from three independent experiments and each experiment was performed in duplicate.

**Soft agar colony formation assay**. Anchorage-independent growth was assessed by plating $0.8 \times 10^4$ cells in 2 ml of 0.6% Noble agar agarose (Difco, USA) over a base of 1% Noble agar in six-well plates. Cells were allowed to grow for 4–6 weeks, during which the media was replaced every week, and colonies formed were visualized by adding 0.015% neutral red in PBS to the media. Images were captured using a custom TE2000-V inverted microscopes with a digital camera (Nikon, UK). The data were obtained from three independent experiments and each experiment was performed in triplicate.

**rOPCML in tumor spheroids invasion assay in Matrigel®**. Five thousand SKOV3 cells were seeded in triplicate in ultra-low attachment U bottom 96-well plates and cultured in RPMI supplemented with 10% FCS at 37 °C, 5% CO2. After 5 days the spheroids were formed and the medium was reduced of growth factors through a series of passages in RPMI deprived of FCS supplement. The spheroids were serum starved overnight, treated with 40 µg ml$^{-1}$ of WT rOPCML or rOPCMLs carrying the R65L, N70H and P95R mutations produced from baculovirus infected insect cells, and then stimulated with 10% FCS. PBS used for the buffer exchange in the final steps of rOPCML purification was used in control to identify any off-target effect due to rOPCML preparation. After 3 h, Matrigel® reduced of growth factor and without phenol red was added to the culture at a 1:1 ratio with the medium. Spheroids were allowed to invade the matrix and images were taken with the Nikon Eclipse TE2000 digital light microscope at 0 h and 72 h at ×4 magnification. Images were analysed with Image J software, and the measurement were expressed as a ratio of invading area over the spheroid area. The data were obtained from two independent experiments and each experiment was performed in triplicate.

**Kinetic stimulation studies with GAS6 and FGF1**. The cell lines were seeded to attain 70% confluence, serum starved overnight and activated with 400 ng mL$^{-1}$ GAS6 (#885-GS, R&D Systems) or 10 ng mL$^{-1}$ FGF1 for 0, 30 min and 3 h and lysates were collected for western blot analysis.

**DUOLink proximity ligation amplification (PLA) assay**. For PLA assay (DUO-Link, OLink Biosciences, Sigma-Aldrich #DUO92102, #DUO92104), the various cell lines were seeded on 13 mm glass coverslips. After the required treatments, the initial part of the immunofluorescence protocol was followed to label the fixed cells with primary antibodies diluted 1:100 from two different species (e.g., mouse anti-OPCML, and rabbit anti-AXL). The rest of the protocol was followed as per manufacturer's instructions.

**Gap closure assays**. The cell lines were seeded at 100% confluence into culture inserts (ibidi #80209) in a 24-well plate and serum-starved overnight. The insert was removed to generate a 500 µm gap, and cells were stimulated with 400 ng mL$^{-1}$ GAS6 or 10 ng mL$^{-1}$ FGF1. Control cells were not stimulated. The cells were visualized under the microscope 18 h later.

**Mammalian 2-hybrid**. OPCML was cloned in the pM vector downstream of the SV40 promoter and GAL4 DNA binding domain, while the extracellular domains of AXL and FGFR1 were cloned in the pVP16 vector downstream of the SV40 promoter and VP16 activation domain. COS-1 cells were transiently transfected at 80% confluence in 24-well plates for 2–3 h with 50 ng of the pGAL4 Luc reporter, 50 ng of each of the different pM and pVP16 constructs and 50 ng of the β-gal transfection control plasmid using Effectene® Transfection Reagent (Qiagen) at a 1:10 DNA to Effectene reagent ratio. Cells were then washed with 1X PBS and incubated in medium overnight. Cells were collected in 1X Cell Culture Lysis Reagent (CCLR) from the Promega Luciferase Assay System and incubated at −80 ˚C for 30 min. Luciferase activity was measured by adding lysates and Luciferase Assay Substrate in Assay buffer (Promega) in equal volumes to white flat-bottomed 96-well plates (PerkinElmer) followed by reading with a VICTOR™ XLight Luminescence Plate Reader (PerkinElmer) using the emission filter slot A7.

Transfection efficiency was measured using the Galacto-Light Plus™ beta-Galactosidase Reporter Gene Assay System (Applied Biosystems) according to the manufacturer's guidelines. Each experiment was carried out in quadruplicate, and the luciferase measurements were normalized for the β-galactosidase readings and are expressed as mean ± standard error of mean (s.e.m).

**R428 IC50**. Cells were seeded in a 96-well plate at 5000 cells/well and treated the following day with different concentrations of the AXL inhibitor, R428 (Selleckchem). Cell viability was evaluated 48 h later using the CellTiter MTS cell proliferation assay (Promega, USA). The experiment was repeated at least three times with a minimum of three replicates, and IC50 values were calculated through GraphPad Prism software (USA).

**Apoptosis assay**. Cells were seeded and treated as above. One plate was used to assess the levels of apoptosis using the Caspase-Glo® 3/7 Assay (Promega, UK), which measures cleaved Caspase-3 and Caspase-7 activities, whereas the second plate was used to determine viable cells by MTS proliferation assay (Promega, USA). The assay was performed according to manufacturer's guidelines. Briefly, following drug treatment, an equal volume of Caspase-Glo® 3/7 reagent was added to each well and plates were incubated for 1 h at room temperature. Luminescence was measured using a LUMIstar OPTIMA microplate reader (BMG Labtech, UK).

**Proliferation assay**. To study cell proliferation, we used MTS-1 proliferation assay (Promega). Cells were seeded at 2000 cells/well in 100 µl in a 96-well plate, and baseline readings were taken on the same day of the seeding as soon as the cells settled. For baseline and each time-point (6, 24, 48, 72, 96 h), 10 µl of MTS-1 solution was added to each well, incubated for 1 h and absorbance was measured at 490 nm.

**Cell adhesion assay**. Cell adhesion onto Fibronectin, collagen I, collagen IV, Laminin I and Fibrinogen was evaluated using the CytoSelect cell adhesion assay (Cell Biolabs, USA), according to the manufacturer's protocol. Cells were suspended in serum-free media at a concentration of $1 \times 10^6$ cell ml$^{-1}$. A total of 150 µl ($1.5 \times 10^5$ cells) were seeded into each well of a plate pre-coated with different ECM proteins. The plate was incubated at 37 °C for 90 min, after which time the wells were gently washed five times with PBS, followed by staining with cell stain solution (0.09% crystal violet) for 10 min at room temperature. Wells were then washed five times with deionized water, and the plate was left to dry before dissolving cells in 200 µl of cell extraction solution (10% acetic acid) and incubating on an orbital shaker for 10 min. One hundred and fifty microliter of each extract sample was finally transferred into a 96-well plate, and the optical density (OD) was measured at 560 nm in computer-interfaced 96-well tunable microtiter plate reader (OPTImax Molecular Devices).

**Animal welfare**. All the procedures involving mice or chicken embryos were performed in agreement with Community and National legislation (Directive 2010/63/EU; Animals Scientific Procedures Act 1986) and also in agreement with the Basel Declaration. The experimental protocol received prior approval by the competent National Authority (project licence reference: PPL 70/7997) and by the Imperial College's Animal Welfare and Ethical Review Body.

**Mouse model**. Eight-week old athymic female mice were injected intraperitoneally with $5 \times 10^6$ cells in 200 ml of PBS. Mice were sacrificed after 10 weeks by Schedule 1 methods. Ascites was aspirated with a needle. Tumors were removed and weighted.

**Chick chorioallantoic membrane model**. Fertilized chicken eggs were purchased from Henry Stewart & Co. Ltd (Fakenham, UK), cleaned and incubated at 37.5 °C with 50% relative humidity on embryonic day (ED) 0, for implantation on ED9. Under sterile conditions, a small hole was pierced through the egg's taglion using a 19 G needle and 3 mL of albumen were removed to decrease the volume inside the egg and to ensure that the Chick chorioallantoic membrane (CAM) is not damaged when the egg shell is opened. Semi-permeable sterile tape was placed over the top of the egg's shell and sharp angled scissors were used to excise the shell and cut a 2 cm² window to expose the CAM. This window was sealed with Suprasorb® F sterile wound-healing film (Lohmann & Rauscher, Germany). Between ED7 and ED10 of incubation, the membranes were ready for the grafting of cancer cells. On ED9, grafts were prepared by suspending 106 cells in 100 µl of Matrigel (BD Biosciences) and gently deposited onto the membrane. After inoculation, the window was then re-covered and sealed, then the eggs were placed back in the incubator. Tumor growth and viability of the embryo was checked daily. Tumor dimensions were measured using SteREO Discovery.V8 microscope (Zeiss, Germany) with Zen 2.0 blue edition software (Zeiss).

**Data analysis**. Images were compiled in Image J. Data were analysed using GraphPad Prism 7. Results are presented as mean ± s.e.m. unless otherwise indicated. Statistical analysis was performed with two-tailed student's $T$-test as appropriate. A $p$-value < 0.05 was considered significant.

**Reporting summary**. Further information on research design is available in the Nature Research Reporting Summary linked to this article.

## Data availability

The atomic coordinates and structure factors for the crystal structure of the OPCML protein determined in this study have been deposited in the Protein Data Bank with accession code 5UV6 https://www.rcsb.org/structure/5UV6. All other data generated or analyzed during this study are included in this published article (and in its accompanying Supplementary Information). A source data file is available for this manuscript.

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

## Acknowledgements

X-ray diffraction experiments used resources of the Advanced Photon Source, a U.S. Department of Energy (DOE) Office of Science User Facility operated for the DOE Office of Science by Argonne National Laboratory under Contract No. DE-AC02-06CH11357. Use of the Lilly Research Laboratories Collaborative Access Team (LRL-CAT) beamline at Sector 31 of the Advanced Photon Source was provided by Eli Lilly Company, which operates the facility. Small angle X-ray scattering research used beamline 16-ID of the National Synchrotron Light Source II, a U.S. Department of Energy (DOE) Office of Science User Facility operated for the DOE Office of Science by Brookhaven National Laboratory under Contract No. DE-SC0012704. This project was funded by Ovarian Cancer Action and supported by Imperial College Biomedical Research Centre, Cancer Research UK Clinical Centre and Experimental Cancer Medicine Centre and by the US National Institutes of Health grant AI38996 (LJS). M.A. is funded by the King Abdulaziz City for Science and Technology (KACST) scholarship. A.T.M. is funded by Fundação para a Ciência e a Tecnologia (FCT, Portugal, PhD grant reference: SFRH/BD/92191/2013). NC acknowledges support from Opticryst, a European VI Framework Program, Project LSHG-CT-2006-037793. We thank Janelle Hayes (UMass Medical School) for assistance with the SEC-MALLS measurements. We thank Kay Diederichs (University of Konstanz) for useful discussions about translational pseudosymmetry and its impact on diffraction intensities. Initial attempts to overexpress and characterize OPCML in eukaryotes took place at the N.C.S.R. 'Demokritos', Athens, Greece and we thank Emmanuel Saridakis, Irene Mavridis, Efstratios Stratikos and Petros Giastas. We thank Vladimir Pelicic for critical reading of the paper. E.V.N.M. and E.W.T. thank the Engineering and Physical Sciences Research Council of the UK for support through the Imperial College Centre for Doctoral Training (grant EP/L015498/1).

## Author contributions

E.Z. and H.L. undertook bioinformatic analysis of OPCML mutations. J.R.B. purified and analyzed wild-type OPCML and point mutants and crystallized OPCML. J.R.B. and L.J.S. solved the structure and interpreted mutations. Z.M. performed SAXS studies. G.C.W. performed kd measurements. M.A. prepared the lentivirus, selected the transduced cells and performed all the phenotypic experiments. J.A. performed the differential interaction experiments. M.M. and C.V. performed the spheroid invasion assay. M.A. and A.T.M. performed the CAM experiment. M.J. performed expression experiments. E.M. and E.K. cloned the domains. E.W.T. supported E.V.N.M. N.C. supported initial experiments. E.Z. and C.R. supervized the phenotypic experiments. E.Z., L.J.S, C.R., and H.G. directed experimental approaches and analysis, C.R. performed the statistical analysis, the experiments with HCT16 cells, the domains and the in vivo mouse experiment, J.R.B., C.R., L.J.S., and H.G. wrote the paper.

## Additional information

**Competing interests:** H.G. is a VP head of Oncology Clinical Discovery at AstraZeneca (Concurrent with his position as Imperial College London tenured chair in medical oncology) and has ownership interest (including patents) in Patent to develop OPCML-based therapeutics. The remaining authors declare no competing interests.

