## [Peer Review File · Nature Communications]

Reviewers' comments:

Reviewer #1 (Remarks to the Author):

Review for Birtley et. al. "Clinical inactivating mutations of distinct structural domains of the tumor suppressor OPCML reveal new cancer-associated functions"

OPCML is a GPI-anchored glycoprotein with three Ig-like domains that shows tumor suppressor activities, possibly by inactivating certain receptor tyrosine kinases.

Here, Birtley et. al. have determined the crystal structure of intact OPCML, the first structure of a member of the IgLON family, which also includes LSAMP, NEGR1, HNT and IgLON5. In spite of their importance, not much is known about this family of glycoproteins, and the new and exciting results presented by Birtley et. al. will certainly boost future research and achievements in the IgLON field.

The crystal structure reveals a V-shaped dimeric arrangement of OPCML with the membrane distal Ig domain1 (D1) as the homotypic dimerization domain.

The authors mapped somatic cancer-associated mutations (which they have found in the COSMIC and TCGA databases <http://cancer.sanger.ac.uk/cosmic/gene/analysis?ln=OPCML>) onto the crystal structure, and in this way, could focus their further functional investigations on a selected few strategic regions on the receptor surface.

The investigated regions are: the dimeric interface, N-linked glycosylation that probably participates in dimerization, and cancer mutation clusters on D2 and D3.

Probably, the most useful mutation is R65L that turns the receptor monomeric, while keeping its structural integrity. Another interesting mutation is N70H, which eliminates a conserved and most probably functional N-linked glycosylation.

The authors then conducted a wide series of experiments, comparing the biochemical and functional properties and effects of w.t vs. OPCML mutants. Biochemical experiments included SAXS and SEC-MALS for determining oligomerization state in solution, and PLA for detecting interactions with various RTKs on cell surfaces. Experiments in cell culture monitored the effects of ectopically expressed OPCML, alone and in concert with RTKs, on cell migration, invasion, colonization, and adherence to isolated ECM components. Finally, w.t and OPCML mutants were examined in in vivo cancer models.

All the experiments were well-conducted and analyzed, and the manuscript is well written.

I, therefore, support publication in Nat. Communications, given that the following experiments will be performed and comments addressed.

Major points

In several places in the article, including the abstract: "... analyzing a database of tumor sequences, we discovered an array of OPCML somatic missense mutations in various tumor types.", or in page 3 : "We identified a number of patients with somatic mutations in OPCML" . Page 19: "Somatic mutation in clinical cancer has been previously described in only one patient with ovarian cancer (P95R)2." and "This is the first study to identify a large number of clinically occurring somatic mutations in the OPCML gene of cancer patients. " or in the discussion (the worst) "This is the first study to identify a large number of clinically occurring somatic mutations in the OPCML gene of cancer patients." make it sound like the authors have acquired new cancer mutations data, while they have merely used existing data from COSMIC and TCGA. It is vital that the authors will amend this impression, and refrain from using terms such as "discovered" or even "identified".

New experiments warranted:

1. Dimerization constants (apparent Kd) measurements for w.t. OPCML and the R65L mutant, in solution and on cell surfaces.
2. Identification of the RTK interacting domains. Are the interactions direct or ECM mediated? Is

there one mechanism for all RTKs interactions?

3. Binding constants (apparent K_d) measurements for w.t. and mutant proteins with the ECM components.

The dimeric structure looks a lot like the Nectin structure that was reported by Narita et. al. in <https://www.ncbi.nlm.nih.gov/pmc/articles/PMC3069466/>

There are also some functional similarities between Nectins and the IgLON family, i.e. cis and trans interactions. It may be useful to address these similarities from an evolutionary and mechanistic perspectives in the discussion.

Minor Points

Page 20: "Strikingly, many clinical mutations were found to map to the dimerization interface, indicating that the quaternary structure might be essential for OPCML's tumor suppressor function."

I am not convinced that the mutation rate is significantly higher at the dimer interface in comparison to other regions. And if you claim that dimerization is key for IgLON tumor suppressor function, I expect to see a better job showing that.

Page 22: "The finding that D1 comprises 26.9% of the OPCML protein sequence but harbors 38.7% of the mutations suggests that this domain is important for the tumor suppressor properties of the protein."

Again, I don't think this is correct or significant.

N70 Glycosylation is located next to the dimerization interface and N70H mutation has an impact over OPCML activities. While the functional data comes from experiments with mammalian cells, the structural and biochemical information comes from protein produced in insect cells. As the Glycan structures are different between human and insects, the crystal structure is not telling everything – at least not at the N70 surroundings. the authors need to make this point very clear and to address the potential implications of these differences in their discussion.

Figure 1 is missing scale bars – what is the distancing between the membrane proximal domains? What is the distance between the membrane and the membrane distal edge?

How were the buried surface area and solvation free energy gain calculated?

Page 5: "These values are consistent with those expected for a physiologically-relevant dimer." Either provide a reference or refrain from making this statement.

Page 6: Better to include the SEC-MALS molecular weight results (fig. S5) showing a dimer of w.t. protein along with the SAXS results.

Include a (supplementary) figure of crystal packing and detailed figure and written description of all crystal contacts.

Reviewer #2 (Remarks to the Author):

These authors have previously published on the tumor suppressor gene, OPCML, its inactivation by somatic epigenetic methylation and particularly its role in regulating and degrading receptor tyrosine kinases in ovarian cancer. Here they extend this to a detailed analysis of its rare protein coding mutations, which occur across all cancer types, with a frequency from 0.1%, to 3% in

gastric cancer and 5% in melanoma. By solving the structure of OPCML, they determined that certain regions affected by clinical mutations were implicated in the functional tumour phenotypes observed. Specifically, they showed that many clinical mutations mapped to the dimerization interface, indicating that quaternary structure might be essential for OPCML's tumor suppressor function and that even rare single point mutations in human cancers can impair OPCML's tumor suppressive function. These findings are important and have relevance beyond the published work.

The authors claim that this is the first reported crystal structure of a member of the immunoglobulin family. They note a homodimeric arrangement and predict that this arrangement and the quaternary structure, and presumably the predilection for mutations involving the dimerization surface, will be found throughout the IgLON family. This is therefore a finding of some significance beyond just the relevance for OPCML.

Three distinct clinical mutations were studied, representing different types of mutations, showing the importance of interaction of OPCML with RTKs, for its TSG activity. The analysis of one particular mutation, Pro95, revealed that interaction of OPCML with the AXL signalling pathway was particularly important, more so than interaction with FGFR1, particularly in terms of in vivo tumorigenicity.

Three independent ovarian cancer cell lines were analysed in vitro and one cell line (SKOV3 cells) was analysed in vivo in immunodeficient mice and in the chick CAM assay. The in vitro and in vivo data are presented with clarity and the differences are evident. The SKOV3 line, on which most analysis was performed, is known to be a line which has HER2 amplification and a PIK3CA mutation and therefore resembles an RTK-dependent form of HGSOC, rather than the more typical p53-mutant form of HGSOC. All three cell lines have been in culture for decades and hence it is still unclear whether these findings are of direct relevance for fresh unmanipulated cancers, but it is likely that they are, given that the mutations themselves have been found across all cancer types in human tumours. The potential limitation will be that the AXL/GAS6 vs FGFR1 finding may well be context-dependent, with examples of other context-dependent complexity, yet to be discovered. Have other examples been studied / predicted? For this work to have relevance more widely, a tumor context outside ovarian cancer should be studied.

The importance of the integrity of the three domains for OPCML's full tumor suppressor activity in vivo is postulated, at least in these contexts studied, yet it may be possible in other contexts, with other types of mutations that this might not always be the case. Should this statement be qualified?

Reviewer #1 (Remarks to the Author):

Review for Birtley et. al. "Clinical inactivating mutations of distinct structural domains of the tumor suppressor OPCML reveal new cancer-associated functions"

OPCML is a GPI-anchored glycoprotein with three Ig-like domains that shows tumor suppressor activities, possibly by inactivating certain receptor tyrosine kinases. Here, Birtley et. al. have determined the crystal structure of intact OPCML, the first structure of a member of the IgLON family, which also includes LSAMP, NEGR1, HNT and IgLON5. In spite of their importance, not much is known about this family of glycoproteins, and the new and exciting results presented by Birtley et. al. will certainly boost future research and achievements in the IgLON field.

The crystal structure reveals a V-shaped dimeric arrangement of OPCML with the membrane distal Ig domain1 (D1) as the homotypic dimerization domain.

The authors mapped somatic cancer-associated mutations (which they have found in the COSMIC and TCGA databases <http://cancer.sanger.ac.uk/cosmic/gene/analysis?ln=OPCML>) onto the crystal structure, and in this way, could focus their further functional investigations on a selected few strategic regions on the receptor surface.

The investigated regions are: the dimeric interface, N-linked glycosylation that probably participates in dimerization, and cancer mutation clusters on D2 and D3.

Probably, the most useful mutation is R65L that turns the receptor monomeric, while keeping its structural integrity. Another interesting mutation is N70H, which eliminates a conserved and most probably functional N-linked glycosylation.

The authors then conducted a wide series of experiments, comparing the biochemical and functional properties and effects of w.t vs. OPCML mutants. Biochemical experiments included SAXS and SEC-MALS for determining oligomerization state in solution, and PLA for detecting interactions with various RTKs on cell surfaces. Experiments in cell culture monitored the effects of ectopically expressed OPCML, alone and in concert with RTKs, on cell migration, invasion, colonization, and adherence to isolated ECM components. Finally, w.t and OPCML mutants were examined in in vivo cancer models.

All the experiments were well-conducted and analyzed, and the manuscript is well written.

I, therefore, support publication in Nat. Communications, given that the following experiments will be performed and comments addressed.

Major points

In several places in the article, including the abstract: "... analyzing a database of tumor sequences, we discovered an array of OPCML somatic missense mutations in various tumor types.", or in page 3 : "We identified a number of patients with somatic mutations in OPCML" . Page 19: "Somatic mutation in clinical cancer has been previously described in only one patient with ovarian cancer (P95R)²." and "This is the first study to identify a large number of clinically occurring somatic mutations in the OPCML gene of cancer patients. " or in the discussion (the worst) "This is the first study to identify a large number of clinically occurring somatic mutations in the OPCML gene of cancer patients." make it sound like the authors have acquired new cancer mutations data, while they have merely used existing

data from COSMIC and TCGA. It is vital that the authors will amend this impression, and refrain from using terms such as “discovered” or even “identified”.

We apologise for giving the wrong impression and we thank the reviewer for highlighting this. We have rephrased our text accordingly.

“we analysed a database of tumor sequences, and uncovered OPCML somatic missense mutations from various tumor types” has been rephrased on page 2 to *‘By analysis of databases of tumor sequences, we found OPCML somatic missense mutations from various tumor types’*.

“We identified a number of patients with somatic mutations...” has been rephrased on page 3 to *“A number of patients presented exhibited somatic mutations of OPCML...”*

“Somatic mutation in clinical cancer has been previously described in only one patient with ovarian cancer (P95R)².” has been deleted from the discussion.

“This is the first study to identify a large number of clinically occurring somatic mutations in the OPCML gene of cancer patients.” was deleted from the discussion.

New experiments warranted:

1. Dimerization constants (apparent K_d) measurements for w.t. OPCML and the R65L mutant, in solution and on cell surfaces.

As requested, we have determined dimerization constants in solution for both WT and R65L proteins, using a gel-filtration assay. As expected, the value for WT (K_d = 1.1± 0.3 μM) was consistent with dimerization under physiologic conditions. We estimated a lower limit on the value for R65L (K_d = >270 μM), based on detection limits in our assay and concentrations used. The new data are shown in new data panels (Supplementary Figure 6C and 6D).

Supplementary Figure 6. Biophysical characterization of WT and point-mutated OPCML. WT and P95R are predominantly dimeric in solution and R65L is predominately monomeric. (A) Gel filtration analysis of WT and OPCML mutants. The elution profiles of WT (red), P95R (blue), R65L (black) and N70H (green) is shown along with molecular weight size standards (in kDa). WT and P95R elute similarly, whilst R65L elutes more slowly. N70H is aggregated. **(B)** The molecular masses of WT, P95R, R65L and N70H (shown in red, blue and black, respectively) were calculated from the elution profile given by SEC-MALS. The calculated masses for WT was 67.3 ± 0.2 kDa, P95R 70.1 ± 0.2 kDa and R65L 36.2 ± 0.1 kDa. OPCML has a molecular mass of approximately 31.7 kDa, as calculated from the primary sequence. **(C)** Concentration dependence of gel filtration profiles for WT (20 nM to 32 μ M) and R65L (1 to 32 μ M). **(D)** Estimation of monomer-dimer dissociation constant K_d . Peak positions from panel C (circles for WT and triangles for R65L) were fit to an equation describing the monomer-dimer equilibrium (see Methods) showing fit values for WT (95% confidence interval 0.565 to 1.997 μ M) and R65L (840 ± 630 μ M, 95% confidence interval lower limit 270 μ M with no upper limit determined).

We added text to the manuscript to describe this result, in the main text:

“we found it [i.e. R65L] to be predominantly a monomer in solution at concentrations up to 32 μ M (Supplementary Figure 6C), with a K_d at least 250-fold weaker than for WT OPCML (Supplementary Figure 6D)”

and in the methods section:

“To determine the binding affinity of OPCML dimerization, various concentrations of WT and

R65L OPCML were subjected to size exclusion chromatography. Briefly, dilutions of purified soluble WT and R65L OPCML were prepared and allowed to equilibrate for 24 hours. Fifty microliters of each dilution were run over a Superdex 200 10/30 GL column at 0.5 mL min⁻¹. The resulting chromatograms were normalized to account for small baseline shifts and slight differences in injection volume. Peak elution volumes at each dilution were plotted against concentration and fit to the equation $EV_{obs} = EV_{monomer} + (1 - fractMonomer)(EV_{dimer} - EV_{monomer})$, where EV_{obs} is the elution volume observed at particular total protein concentration P_{tot} , $EV_{monomer}$, and EV_{dimer} are fit values for monomer and dimer elution volumes, respectively, and $fractMonomer$ is fraction of the total protein concentration represented by the monomer, determined using the formula $fractMonomer = (-K_d + \sqrt{K_d^2 + 8 * P_{tot} * K_d}) / 4P_{tot}$, where K_d is the fit value for the dimerization constant³⁴. For estimation of a lower bound on the R65L OPCML K_d , EV_{dim} was constrained to the same value as for WT OPCML. GraphPad Prism 7 software (USA) was used for curve fitting.”

We are not aware of an analytical method to measure dissociation constants on the cell surface, but we were able to evaluate the oligomerization state of the full-length protein as present on the cell surface by extracting the membrane proteins and by western blot. These results demonstrate that indeed the wild type dimeric and the R65L mutant monomeric forms of OPCML exist also in physiological conditions in cells. These new results are shown in Figure 3A, right panel:

Figure 3. R65L, N70H and P95R mutants show loss of their tumor suppressor function in vitro. ...Samples in the right panel in A were run without heat denaturation...

with the corresponding text added to the manuscript:

“Furthermore, when WT and the R65L mutant were extracted from transduced cells and analysed by SDS-PAGE without prior heat denaturation, WT ran as a dimer while R65L ran exclusively as a monomer (Figure 3A), demonstrating that WT OPCML exists as a dimer also in cells and that the R65L mutation does indeed disrupt dimerization.”

2. Identification of the RTK interacting domains. Are the interactions direct or ECM mediated? Is there one mechanism for all RTKs interactions?

To address this point, we expressed domains D1, D2 and D3 separately (with the signal sequence for secretion at the N-terminal and the GPI anchor at the C-terminal) in cancer cells and studied their interaction with AXL, which is one of the best characterised partners of OPCML. We found the domains to be correctly expressed and localised at the plasma membrane in SKOV3 cells (Supplementary Figure 9A). The association with AXL was investigated by DuoLink and, as shown in Supplementary Figure 9B, we found that AXL interacts preferentially with domain 1, which is also where the P95R inactivating mutation is localised. Interestingly, this interaction is also promoted by AXL binding to GAS6 (Supplementary Figure 9C), similarly to what we have observed with full-length OPCML, while this is not the case for the other domains.

Supplementary Figure 9. OPCML interacts with AXL mainly via the D1 domain. (A) SKOV3 cells were stably transduced with D1, D2 or D3, which are all HA-tagged, and stained in red with an anti-HA antibody. Nuclei are stained by DAPI (cyan). Scale bar = 20 μm. (B) Cells were grown in full medium and the interaction between the different domains of OPCML and AXL was measured by DuoLink. The number of dots was quantified and graph shows the mean ± s.e.m of 3 independent experiments. (C) Cells were starved and then stimulated with Gas6 for 3h. The interaction between OPCML's domains and AXL was measured by DuoLink. The total area covered by the signal was quantified and the graph shows the mean ± s.e.m of 3 independent experiments. Student t-test: * $p < 0.05$.

However, since RTKs have very different amino acid sequences and 3D structures, we doubt that this mechanism of association would be universal for all OPCML/RTKs interactions, as shown for example by the fact that the P95R mutation does not disrupt the binding to FGFR1 (Figure 4).

Regarding the ECM role, the association of WT OPCML with AXL and FGFR1 and the differential interaction between the P95R mutant and AXL and FGFR1 has also been demonstrated by mammalian 2 hybrid (Figure 4C and D). In this type of assay, the soluble truncated proteins interact in the cytosol of the transfected cells and not on the plasma membrane. Under these conditions, they are not exposed to secreted extracellular matrix components. Therefore, this indicates that ECM components are not required for, or at least do not play a major role in the interaction of OPCML with the RTKs.

3. Binding constants (apparent Kd) measurements for w.t. and mutant proteins with the ECM components.

In order to test the hypothesis that OPCML could interact directly with ECM components, we assessed by Biacore the binding of WT OPCML to collagen I and IV. We chose these two substrates as we have observed that cells that express OPCML have higher adherence to these (Figure 7B and C). Specifically, collagen I and IV were immobilized on the surface of the Biacore chip and WT OPCML in varying concentrations up to 50 μ M was tested for binding. However, no detectable binding interaction was observed (data not shown). These results indicate that the effect of OPCML on cell adherence to these substrates may be indirect and mediated by intermediate partners.

We have modified the text of the manuscript to remove the ambiguity that OPCML binds directly to the ECM and the sentence *“These results show an important role for D1 residues in the interaction with the extracellular matrix component collagen I.”* has been removed.

The dimeric structure looks a lot like the Nectin structure that was reported by Narita et. al. in <https://www.ncbi.nlm.nih.gov/pmc/articles/PMC3069466/>

There are also some functional similarities between Nectins and the IgLON family, i.e. cis and trans interactions. It may be useful to address these similarities from an evolutionary and mechanistic perspectives in the discussion.

We have now inserted the reference to Nectins given above in our manuscript (reference 38) and we have included the following text in the discussion:

“The quaternary structure of the OPCML homodimer presented here looks highly similar topologically to that of nectins³⁸. Nectins are composed of 3 consecutive Ig-like domains, are also plasma-membrane associated and can mediate cell-cell communication via pairwise cis- and trans- interaction homo- or heterotypically with other nectin family members. The IgLON family of proteins have been suggested to pair in a similar fashion^{8,9} and for these reasons a possible common mechanism could be suggested.”

Minor Points

Page 20: *“Strikingly, many clinical mutations were found to map to the dimerization interface, indicating that the quaternary structure might be essential for OPCML’s tumor suppressor function.”*

I am not convinced that the mutation rate is significantly higher at the dimer interface in comparison to other regions. And if you claim that dimerization is key for IgLON tumor suppressor function, I expect to see a better job showing that.

The above sentence has been modified in the manuscript as outlined below:

We originally had “ Strikingly, many clinical mutations were found to map to the dimerization interface, indicating that the quaternary structure might be essential for OPCML’s tumor suppressor function. To test this hypothesis, we characterized a protein with the R65L clinical mutation...”

We now put “With a large proportion of mutations being present in D1 and with dimerization being mediated by this domain, we hypothesized an important role for D1 in

mediating OPCML's tumor suppressor functions. To test this hypothesis, we characterized a protein with the R65L clinical mutation...".

Page 22: "The finding that D1 comprises 26.9% of the OPCML protein sequence but harbors 38.7% of the mutations suggests that this domain is important for the tumor suppressor properties of the protein."

Again, I don't think this is correct or significant.

The above sentence has been deleted from the manuscript.

N70 Glycosylation is located next to the dimerization interface and N70H mutation has an impact over OPCML activities. While the functional data comes from experiments with mammalian cells, the structural and biochemical information comes from protein produced in insect cells. As the Glycan structures are different between human and insects, the crystal structure is not telling everything – at least not at the N70 surroundings. the authors need to make this point very clear and to address the potential implications of these differences in their discussion.

The following text has been inserted into the discussion:

"The crystal structure of OPCML presented here came from insect cell-expressed material but much of the functional data employed OPCML from a mammalian source. It is possible that interactions between glycan at asparagine 70 and protein at the D1-D1 dimer interface could be different depending on the origin of the material. Insect cell and human N-linked glycans typically differ at terminal sialylation sites (present in human not insect) and occasionally by addition of α 1,3-linked fucose at the initial GlcNac³⁷. We did not observe ordered density extending the terminal glycans for any of the glycans modeled, nor did we observed α 1,3-linked fucose at the initial GlcNac. Of note, when the recombinant proteins produced in insect cells were tested in phenotypic assays (spheroid invasion, Fig 3F), they showed a loss of function similar to the mutant proteins produced directly by the cells. These data indicate a strong similarity in terms of functionality between insect- and mammalian-produced OPCML proteins."

Figure 1 is missing scale bars – what is the distancing between the membrane proximal domains? What is the distance between the membrane and the membrane distal edge?

Distances have been measured with the assistance of COOT and these have been added to Figure 1A. They are mentioned in the manuscript:

"Based on the quaternary structure seen in the asymmetric unit, the dimer would be anchored into the plasma membrane by two D3-linked GPI anchors (Figure 1A). The D3 C-terminal ends are located approximately 180 Å apart, and thus the top of the D1 dimerization interface could extend above the plane of the plasma membrane by approximately 73 Å (Figure 1A)."

Figure 1. Crystal structure of the OPCML homodimer. (A) Ribbon representation overlaid on a transparent surface illustrating the dimeric architecture of OPCML. One monomer is colored blue and the other wheat. N-linked glycosylation (on asparagines 70, 293 and 306) are shown as orange spheres. Location of P95 indicated by *. N- and C- termini are indicated. Red parallel lines show the relative orientation of OPCML to the plasma membrane (on the extracellular side). Scale bars are shown to illustrate maximum dimensions.

How were the buried surface area and solvation free energy gain calculated?

These values were calculated using PISA and this has now been referenced in the text:

“This interface comprises 874 Å² of buried surface area and has a solvation free energy gain (ΔiG) on formation calculated by PISA to be -9.5 kcal M^{-1} .”

Page 5: “These values are consistent with those expected for a physiologically-relevant dimer.” Either provide a reference or refrain from making this statement.

The above statement was deleted from the text as follows:

We previously put “This interface comprises 877 Å² of buried surface area and has a solvation free energy gain (ΔiG) on formation calculated to be -9.5 kcal/M . These values are consistent with those expected for a physiologically-relevant dimer. These values are larger than for other intermolecular interaction sites observed in the crystal, the largest of which represents crystal contacts mediated by the C-terminal end of one molecule inserting between the B and G strands of D1 in another with 554 Å² buried surface area and solvation free energy gain of -6.9 kcal/M .”

We now have “This interface comprises 874 Å² of buried surface area and has a solvation free energy gain (ΔiG) on formation calculated by PISA33 to be -9.5 kcal M^{-1} . These values are larger than for other intermolecular interaction sites observed in the crystal, the next

largest being crystal contacts mediated by the C-terminal end of one molecule inserting between the B and G strands of D1 in another with 554 Å² buried surface area and solvation free energy gain of -6.9 kcal M⁻¹ (Supplementary table 1 and Supplementary Figure 5).”

Page 6: Better to include the SEC-MALS molecular weight results (fig. S5) showing a dimer of w.t. protein along with the SAXS results.

We moved the SAXS data (creating main Figure 1F) in order for it to be alongside the other SAXS data results and a dimer of OPCML.

Figure 1. Crystal structure of the OPCML homodimer. (A) Ribbon representation overlaid on a transparent surface illustrating the dimeric architecture of OPCML. One monomer is colored blue and the other wheat. N-linked glycosylation (on asparagines 70, 293 and 306) are shown as orange spheres. Location of P95 indicated by *. N- and C- termini are indicated. Red parallel lines show the relative orientation of OPCML to the plasma membrane (on the extracellular side). Scale bars are shown to illustrate maximum dimensions. (B) Represented the same as in a) but viewed from the top. The dashed rectangle represents key residues at the dimerization interface and these are shown in detail in C. (C) The D1-D1 homodimerization interface. Arg 65 from one monomer forms a salt bridge with Asp of the other monomer. Below Arg 65 is a stacking interaction involving Trp 82, Ile 74, Leu 69 and Thr 114. Same color scheme as in a) and b). The alpha carbon backbone is shown as thin lines. (D) Stick representation of the β -turn encompassing residue P95, shown in wheat. Residues 92-94 and 97-99 form part of the D and E β -strands respectively. Hydrogen bonds are indicated by dashed lines. (E) Small angle X-ray scattering analysis of WT OPCML. Scattering curves were measured and compared to calculated data of the dimer as seen in the crystal structure and of a monomeric model of OPCML. The fit to the dimer ($\chi = 0.50$) matches more closely than to the monomer ($\chi = 2.35$). (F) Pair-distance distribution function curve for WT OPCML. The curve intercepts the x-axis at $R = 189 \text{ \AA}$, indicating the maximum atomic distance in a single scattering particle. This model-independent parameter is consistent with the maximum manually measured dimension of the dimer structure (approximately 180 \AA), and is significantly larger than the maximum dimension of the monomer structure (approximately 124 \AA).

Include a (supplementary) figure of crystal packing and detailed figure and written description of all crystal contacts.

These have been created with a crystal packing figure (new Supplementary figure 5) and list of crystal contacts (Supplementary table 1) now included with the manuscript.

Supplementary Figure 5. Packing of OPCML in the crystal lattice. Orthogonal views of OPCML packing interactions as seen in the crystal lattice. The unit cell is shown in both and the OPCML peptide backbone shown as a trace.

Interface	Interface area (Å ²)	vation free energy gain (kcal/mol)	Symmetry operator	Chain	Residue number/atom	Distance (Å)	Chain	Residue number/atom
1	873.9	-9.5	x,y,z	Hydrogen bond				
				B	THR 114[OG1]	3.78	A	SER 72[OG]
				B	ARG 65[NH2]	2.96	A	ASP 80[OD1]
				B	SER 72[OG]	3.72	A	THR 114[OG1]
				B	ILE 74[N]	3.76	A	SER 116[OG]
				B	SER 72[OG]	2.98	A	ARG 127[NH2]
				B	ASP 80[OD1]	2.68	A	ARG 65[NH2]
				B	SER 116[OG]	3.87	A	ILE 74[N]
				Salt bridge				
				B	ARG 65[NE]	3.77	A	ASP 80[OD1]
				B	ARG 65[NH2]	2.96	A	ASP 80[OD1]
				B	ARG 65[NH1]	3.02	A	ASP 80[OD2]
				B	ARG 65[NE]	3.61	A	ASP 80[OD2]
				B	ARG 65[NH2]	3.09	A	ASP 80[OD2]
				B	ASP 80[OD1]	3.61	A	ARG 65[NE]
				B	ASP 80[OD1]	2.68	A	ARG 65[NH2]
				B	ASP 80[OD2]	3.70	A	ARG 65[NE]
				B	ASP 80[OD2]	2.92	A	ARG 65[NH2]
				B	ASP 80[OD2]	2.94	A	ARG 65[NH1]
2	553	-6.8	y-1,x,-z+2	Hydrogen bond				
				B	SER 235[OG]	2.96	A	ASN 44[O]
				B	ALA 316[N]	2.91	A	ARG 56[O]
				B	ALA 317[N]	3.07	A	SER 126[OG]
				B	LEU 318[N]	3.17	A	THR 58[O]
				B	SER 235[OG]	3.17	A	THR 46[N]
				B	GLN 238[OE1]	3.68	A	GLN 133[NE2]
				B	ALA 316[O]	2.89	A	THR 58[N]
B	LEU 318[O]	3.22	A	ASP 60[N]				
3	511.2	-8.6	y,x-1,-z+2	Hydrogen bond				
				B	ASN 44[ND2]	3.65	A	SER 235[OG]
				B	MET 42[N]	3.79	A	GLY 315[O]
				B	THR 58[N]	3.06	A	ALA 316[O]
				B	ASP 60[N]	3.57	A	LEU 318[O]
				B	ARG 56[O]	3.05	A	ALA 316[N]
				B	SER 126[OG]	3.04	A	ALA 317[N]
B	THR 58[O]	3.12	A	LEU 318[N]				
4	394.3	-0.8	-y+1/2,x+1/2,z+1/4	Hydrogen bond				
				B	ASN 93[N]	2.93	B	ILE 226[O]
				B	ARG 56[NH1]	3.47	B	ASN 306[O]
				B	ARG 56[NH2]	3.42	B	ASN 306[O]
				B	ARG 56[NH2]	3.05	B	ASN 306[OD1]
				B	LEU 91[O]	2.89	B	THR 305[OG1]
B	LEU 91[O]	2.89	B	ILE 226[N]				
5	368.7	-2	-y+1/2,x-1/2,z+1/4	Hydrogen bond				
				A	ARG 56[NH1]	3.52	A	ASN 306[O]
				A	ARG 56[NH2]	3.32	A	ASN 306[O]
				A	ARG 56[NH2]	3.01	A	ASN 306[OD1]
				A	ASN 93[N]	2.87	A	ILE 226[O]
				A	LEU 91[O]	2.81	A	ILE 226[N]
A	LEU 91[O]	2.88	A	THR 305[OG1]				
6	235.7	-1.6	Hydrogen bond					
			B	LYS 258[N2]	2.87	A	GLU 259[OE1]	
			Salt bridge					
B	LYS 258[N2]	2.87	A	GLU 259[OE1]				

Supplementary Table 1. Shown are the 6 largest contact areas as seen in the OPCML crystal structure and the buried surface area along with the contact residues and distances involved are listed, within 4 Å. Figures were calculated using PISA. The largest contact surface corresponds to the D1-D1 homodimerization interface and contains 874 Å² of interface area. The next 5 largest sites comprise consecutively less surface area and fewer amino acid contact sites.

Reviewer #2 (Remarks to the Author):

These authors have previously published on the tumor suppressor gene, OPCML, its inactivation by somatic epigenetic methylation and particularly its role in regulating and degrading receptor tyrosine kinases in ovarian cancer. Here they extend this to a detailed analysis of its rare protein coding mutations, which occur across all cancer types, with a frequency from 0.1%, to 3% in gastric cancer and 5% in melanoma. By solving the structure of OPCML, they determined that certain regions affected by clinical mutations were implicated in the functional tumour phenotypes observed. Specifically, they showed that many clinical mutations mapped to the dimerization interface, indicating that quaternary structure might be essential for OPCML's tumor suppressor function and that even rare single point mutations in human cancers can impair OPCML's tumor suppressive function. These findings are important and have relevance beyond the published work.

The authors claim that this is the first reported crystal structure of a member of the immunoglobulin family. They note a homodimeric arrangement and predict that this arrangement and the quaternary structure, and presumably the predilection for mutations involving the dimerization surface, will be found throughout the IgLON family. This is therefore a finding of some significance beyond just the relevance for OPCML.

Three distinct clinical mutations were studied, representing different types of mutations, showing the importance of interaction of OPCML with RTKs, for its TSG activity. The analysis of one particular mutation, Pro95, revealed that interaction of OPCML with the AXL signalling pathway was particularly important, more so than interaction with FGFR1, particularly in terms of in vivo tumorigenicity.

Three independent ovarian cancer cell lines were analysed in vitro and one cell line (SKOV3 cells) was analysed in vivo in immunodeficient mice and in the chick CAM assay. The in vitro and in vivo data are presented with clarity and the differences are evident. The SKOV3 line, on which most analysis was performed, is known to be a line which has HER2 amplification and a PIK3CA mutation and therefore resembles an RTK-dependent form of HGSOc, rather than the more typical p53-mutant form of HGSOc. All three cell lines have been in culture for decades and hence it is still unclear whether these findings are of direct relevance for fresh unmanipulated cancers, but it is likely that they are, given that the mutations themselves have been found across all cancer types in human tumours. The potential limitation will be that the AXL/GAS6 vs FGFR1 finding may well be context-dependent, with examples of other context-dependent complexity, yet to be discovered. Have other examples been studied / predicted? For this work to have relevance more widely, a tumor context outside ovarian cancer should be studied.

To answer this question, we investigated the effect of the mutations in another tumor context, i.e. colorectal cancer. We transduced representative mutants from the three domains (R65L, R71C and P95R for D1, E201Q for D2, M278I for D3) into the colorectal cancer cell line HCT116. The mutants were correctly expressed and localised (Supplementary Figure 8 A and B) and they all showed loss of ability to reduce the activation

of the ERK pathway compared to wild type OPCML (Supplementary Figure 8B), thus demonstrating a conserved function in another tumor context.

Supplementary Figure 8. OPCML mutants show impaired tumour suppressor functions in HCT116 cells. (A) Colorectal HCT116 cells were stably transduced with the indicated constructs and protein expression was verified by immunofluorescence with an anti-OPCML antibody (red). Cell nuclei were stained with DAPI (cyan). Scale bar = 20 μ m. (B) Cells were grown in full medium conditions and their signaling analyzed by western blotting with the indicated antibodies.

The following text has been added:

“Similarly, expression of WT OPCML in the colorectal cancer cell line HCT116 (Supplementary Figure 8A) decreased the phosphorylation of ERK1/2 but the OPCML mutants [R65L, N70H, P95R] could not (Supplementary Figure 8B).”

“Furthermore, it [R71C] was unable to block ERK1/2 activation in HCT116 colorectal cancer cells (Supplementary Figure 8B).”

“Similarly, the E201Q and the M278I mutants showed loss of function in HCT116 cells (Supplementary Figure 8B).”

Furthermore, when colorectal cancer cells expressing wild type OPCML were stimulated with GAS6, they showed a strong increase in the level of OPCML/AXL interaction as shown by DuoLink (Supplementary Figure 8C). Similar to previous observations in ovarian cancer cells, the R65L and P95R mutants demonstrated a complete loss of response to GAS6 and the levels of OPCML association with AXL remained very low in the presence of the ligand. These data indicate that the functional interaction between AXL and OPCML is conserved in other cancer types and contexts.

Supplementary Figure 8. OPCML mutants show impaired tumour suppressor functions in HCT116 cells. (C) Cells were starved and then stimulated for 3h with Gas6, the interaction of OPCML WT, R65L and P95R with AXL was measured by DuoLink. The graph shows the mean \pm s.e.m of 3 independent experiments. Student t-test ** $p < 0.01$.

The following text has been added:

“Similar results were obtained also in HCT116 cells, where the addition of Gas6 stimulated the binding of AXL to WT OPCML but not to the R65L and P95R mutants (Supplementary Figure 8C).”

The importance of the integrity of the three domains for OPCML’s full tumor suppressor activity *in vivo* is postulated, at least in these contexts studied, yet it may be possible in other contexts, with other types of mutations that this might not always be the case. Should this statement be qualified?

We agree with the reviewer’s comments. Though all the mutations tested here conferred a loss-of-function phenotype *in vivo* in ovarian cancer cells, it is possible that some of the 287 mutations identified in the databases could actually impair only a subset of functions and not *in vivo* tumorigenicity in certain types of cancers. Unfortunately, it would be extremely complex to test all the 287 mutations in a large number of different types of cancers *in vivo* in order to fully address this important point.

We have modified the discussion to highlight this possibility and we thank the reviewer for noticing it.

REVIEWERS' COMMENTS:

Reviewer #1 (Remarks to the Author):

Birtley et al. have responded quite comprehensively to the comments on their interesting manuscript from the previous round of review, and have largely done so in a convincing way. The paper now seems acceptable for Nature Comm.

Reviewer #2 (Remarks to the Author):

The authors have addressed my concerns. Whilst I would always have preferred that experiments be performed in newer models of the relevant tumour type, for example, fresh organoids or PDX of high grade serous ovarian cancer, switching cancer type to colorectal cancer and finding the same biology, of OPCML mutants showing impaired TSG function, is in keeping with their hypothesis.